# National Institutes of Health research project grant inflation 1998 to 2021

Michael S Lauer[1]*, Joy Wang[2], Deepshikha Roychowdhury[2]

[1]National Institutes of Health Office of the Director, Bethesda, United States; [2]National Institutes of Health Office of Extramural Research, Bethesda, United States

**Abstract** We analyzed changes in total costs for National Institutes of Health (NIH) awarded Research Project Grants (RPGs) issued from fiscal years (FYs) 1998 to 2021 . Costs are measured in 'nominal' terms, meaning exactly as stated, or in 'real' terms, meaning after adjustment for inflation. The NIH uses a data-driven price index – the Biomedical Research and Development Price Index (BRDPI) – to account for inflation, enabling assessment of changes in real (that is, BRDPI-adjusted) costs over time. The BRDPI was higher than the general inflation rate from FY1998 until FY2012; since then the BRDPI has been similar to the general inflation rate likely due to caps on senior faculty salary support. Despite increases in nominal costs, recent years have seen increases in the absolute numbers of RPG and R01 awards. Real average and median RPG costs increased during the NIH-doubling (FY1998 to FY2003), decreased after the doubling and have remained relatively stable since. Of note, though, the degree of variation of RPG costs has changed over time, with more marked extremes observed on both higher and lower levels of cost. On both ends of the cost spectrum, the agency is funding a greater proportion of solicited projects, with nearly half of RPG money going toward solicited projects. After adjusting for confounders, we find no independent association of time with BRDPI-adjusted costs; in other words, changes in real costs are largely explained by changes in the composition of the NIH-grant portfolio.

*For correspondence: Michael.Lauer@nih.gov

Competing interest: The authors declare that no competing interests exist.

## Editor's evaluation

Costs for NIH supported research go up each year and it is important to understand whether those costs are greater than those due to overall inflation which recently is rising more than expected. In this paper Lauer and colleagues report that prior to 2012 NIH costs exceeded inflation, but over the last ten years real NIH costs, match inflationary increases, due in part to salary caps on investigators. More of the funded NIH research during this time period is supporting solicited projects.

## Introduction

Inflation, defined by the United States Federal Reserve as 'the increase in the prices of goods and services over time' (*US Federal Reserve, 2022*), has been a longstanding concern in the biomedical research community. (*Mervis, 2015*) Concern has only increased over the past year given the increased rate of inflation in the general economy.

To comprehend the nature of grant costs and trends, we define the following terms:

- Nominal costs: These are costs exactly as stated. Thus, the nominal total cost of a particular grant in FY2021 might be $450,000, meaning that the amounts listed in financial transactions or grant notices would add up to $450,000.
- Real costs: These are costs taking into account inflationary changes over time. Because of inflation $450,000 in FY2002 would have more purchasing power, that is could acquire more goods and services, than $450,000 in FY2021. Real costs are indexed against a specific year so that a comparison of real costs between two different years would reflect a comparison of purchasing

power, not a comparison of costs as stated. We can think of real costs as enabling us to make 'apples to apples' comparisons in costs over different time periods.

- Logarithm-transformed costs: Grant costs followed a highly right-skewed distribution. We therefore sometimes transform costs to a base log of 10 in order to make distributions more symmetrical and less influenced by extreme values (*West, 2022*).

The National Institutes of Health (NIH) issues different type of research and training awards, but by far the most common type is the "Research Project Grant (RPG)"(*NIH, 2022a*) accounting for over half of the NIH budget. (*NIH, 2022b*) Prices for research project grants (RPGs) awarded by the National Institutes of Health (NIH) may increase over time for at least three reasons:

- Background inflation: Increases in prices across the economy due to increases in the money supply and/or economy-wide demand and supply stresses; these are reflected in general price indices, such as the GDP price index (*NIH, 2022c*) and the Consumer Price Index. (*US Bureau of Labor Statistics, 2022*)
- Research-specific inflation: Increases in prices in the biomedical research and development enterprise; these are reported as the Biomedical Research and Development Price Index (or BRDPI). (*NIH, 2022c*) The BRDPI measures changes in the weighted average of the prices of all the inputs (e.g. personnel services, various supplies, and equipment) purchased with the NIH budget to support research. The weights used to construct the index reflect the actual pattern, or proportions, of total NIH expenditures on each of the types of inputs purchased. Theoretically, the annual change in the BRDPI indicates how much NIH expenditures would need to increase, without regard to efficiency gains or changes in government priorities, to maintain NIH-funded research activity at the previous year's level. In this report we refer to inflation-adjusted grants costs as 'real costs' or as 'BRDPI-adjusted costs'.
- Changes in agency purchasing decisions (or compositional effects): We might imagine an automobile-rental firm that starts one year purchasing 10 mid-size sedans. The following year, it might choose to purchase instead 10 luxury mid-size sedans; costs increase not because of background inflation because of the firm's decisions about what it wants to buy. Alternatively, the firm may purchase two large vans, four mid-sized sedans, and four compact cars. Overall and median costs might not change (compared to the baseline of 10 mid-size sedans), but the firm's management will be acutely aware of the costs of the two large vans. Similarly, NIH Institutes and Centers (IC's) may choose to 'puchase' investigator-initiated R01 awards, R01 awards that cost more (e.g. >$500,000 in direct costs) because of use of large animals, or different size awards (program project grants, cooperative agreements, or small exploratory R21 or R03 awards).

We report on the distribution of nominal and inflation-adjusted prices of funded NIH RPGs since FY1998, the year that the NIH budget doubling began. We find that median and mean inflation-adjusted RPG costs have been largely stable since the doubling ended in FY2003, but that there have been changes in the distribution (variance) of costs, which largely reflect compositional effects as agency priorities have shifted over time.

## Results
### Changes in RPG costs and characteristics over time

Most of this report will focus on real (as opposed to nominal) costs of NIH RPG awards, that is total costs per RPG indexed for the FY2021 BRDPI. For context, between FY1998 and FY2021, NIH issued 827,815 RPG awards of at least $25,000 per year (BRDPI-indexed to FY2021). The number of RPGs and Principal Investigators supported on RPGs increased during the NIH budget doubling (from FY1998 to FY2003), decreased gradually between the end of the doubling and FY2015, and increased again with recent NIH budget increases (*Figure 1*, panel A). Similar trends were seen with R01 equivalent awards (*Appendix 1—figure 1*). Between FY1998 and FY2012 the BRDPI was consistently higher than the GDP Price Index (*NIH, 2022c*); after FY2012, when the government imposed lower caps on compensation of extramural investigators, the BRDPI has fallen to the same levels as the GDP Price Index (*Figure 1*, panel B). Both the BRDPI and the GDP Price Index are projected to increase in FY2022, but decrease to close to FY2021 levels over the next 2 years; these projections should be interpreted with caution given recent price volatility linked to the COVID-19 pandemic and supply-chain interruptions (*Figure 1*, panel B) (*NIH, 2022c*).

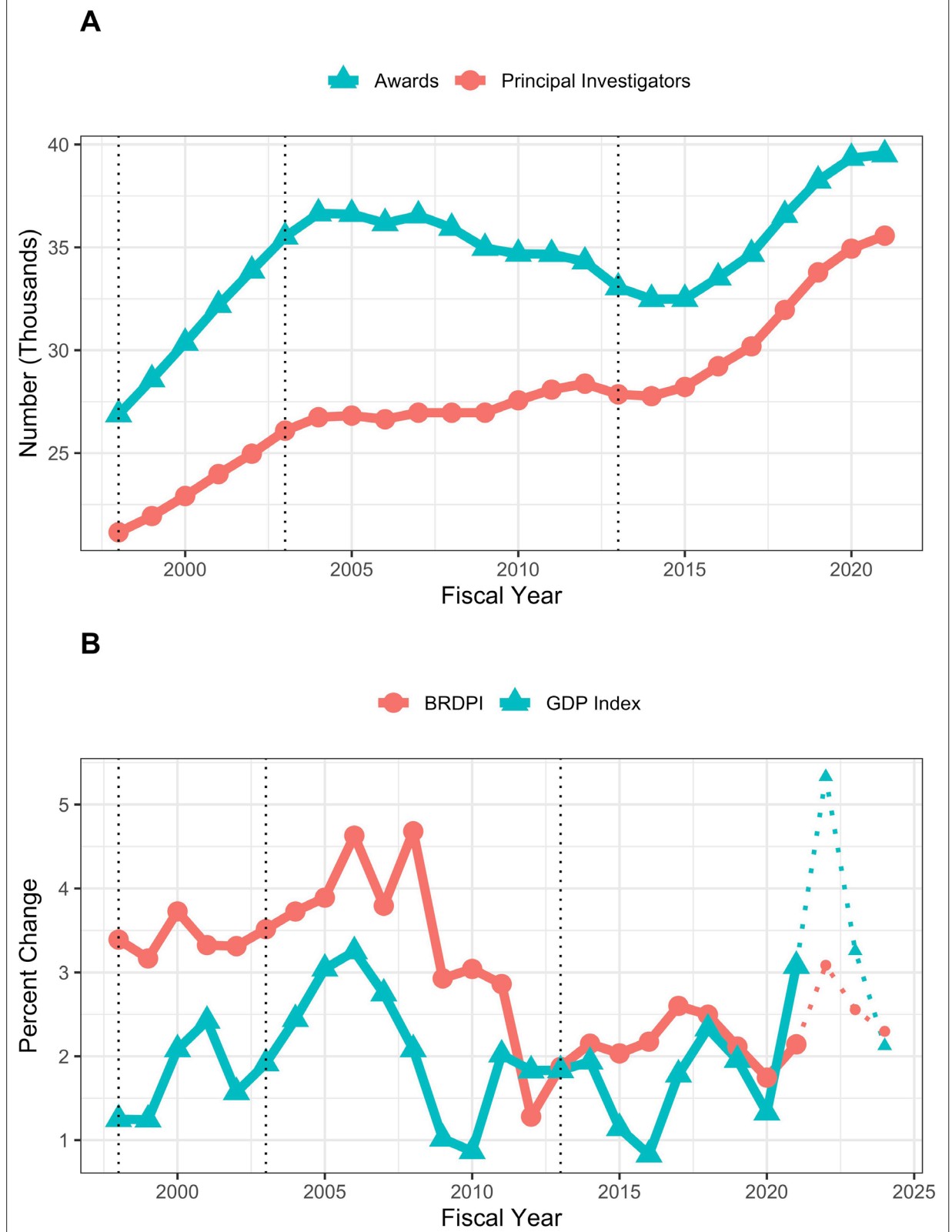

**Figure 1.** Number of funded RPGs and RPG Principal Investigators (panel A) and inflationary indices (panel B) by fiscal year. The vertical dotted lines refer to the beginning and end of the NIH budget-doubling (FY1998 and FY2003) and the year of budget sequestration (FY2013). Indices shown in Panel B for fiscal years 2022, 2023, and 2024 are projected and therefore diplayed with thinner dotted lines.

## Nominal and BRDPI-indexed costs of NIH RPGs over time

Mean and median real (that is FY2021 BRDPI-adjusted) total costs of RPGs increased during the doubling of FY1998-FY2003 (from average values of about $530,000 to about $610,000), fell to a nadir of about $520,000 in FY2013 (the year of budget sequestration), and after a quick rebound in FY2014 has remained relatively stable at about $570,000 since (*Figure 2*). Similar trends were seen with real indirect costs of RPGs, which if anything have increased modestly in more recent year (*Figure 3*) and for total real costs of R01-equivalent awards (*Figure 4*). Indirect costs are not directly linked to the work conducted in a research project and are used to support facilities and administration; some refer to them as overhead (*NIH, 2023a*).

## Characteristics of NIH RPGs over time

Over time there have been decreases in the proportion of unsolicited awards and program ('P') grants, while there have been increases in the proportions of R21 or R03 grants, cooperative agreements, clinical trials (since reliable data were first collected in FY2008), highly expensive projects (defined as those costing at least $5 million in FY2021 BRDPI-adjusted, *that is real*, values), and human-studies only projects (*Table 1*). The proportion of R01-equivalent awards increased during the doubling and then returned to FY1998 levels. Institutes of higher education, independent research organizations, and independent hospitals have consistently accounted for over 97 percent of awards. Among R01-equivalent awards, the proportion of awards with nominal direct costs less than $250,000 has decreased over time, while the proportion of awards with nominal direct costs greater than $500,000 has increased. These values correspond to cut-offs for submission of simplified modular budgets and for required pre-approval for application (*Table 2*).

## Variation in RPG costs over time

We constructed box-plot distributions of FY2021 BRDPI-adjusted total cost per RPG over time (*Appendix 1—figure 2*, *panel A*); these showed means much greater than medians, consistent with highly skewed distributions. The whiskers are also quite long, consistent with fat-tailed distributions. We addressed skewness by log transforming BRDPI-adjusted total costs (*TC*), that is calculating $log_{10}TC_{BRDPI}$. With log-transformation means and medians are nearly equal (eliminating skewness), but the whiskers remain prominent reflective of fat tails on both more expensive and less expensive ends (*Appendix 1—figure 2*, *Panel B*).

Careful inspection of both panels (*Appendix 1—figure 2*) reveals an interesting pattern in variation. From the time of the doubling until about FY2010, the distance between the whisker tips decreased. We call this distance the 'whisker range'. From FY2012 through FY2021 whisker ranges increased, exceeding levels for the doubling for untransformed costs, and not quite reaching doubling levels for log-transformed costs. We can think of the upper (and lower) whisker tips as the most extremely expensive (inexpensive) award that is not an outlier; the distance of the tips from the center (median) reflects the agency's general willingness to vary its funding instruments. *Figure 5* shows the whisker ranges declined from $920,000 to $750,000 between FY2002 and FY2010 and increased to over $1 million in FY2021 (panel A, with log-transformed values shown in panel B).

What might be behind the increasing extremes (higher and lower) over the past 10–15 years? In FY1998, the top centile of RPG awards received 8% of funding, rising to 12% FY2017; this 4% absolute difference means that an additional $850 million were awarded to approximately 350 grants. There was little change in the proportion of funding going to the top decile; thus the upper extreme seems to be driven by increases in funding going to the most expensive awards (*Appendix 1—figure 3*).

## Solicited and unsolicited projects over time

Expensive awards might be linked to agency solicitations. Before FY2010 unsolicited RPGs had a central tendency towards greater costs, but since then solicited awards were more costly (*Figure 6*, panel A). The proportion of solicited projects increased from 20% to 30% from FY1998 to FY2005, then remained stable until FY2016, and increased to 40% from FY2016 to FY2021. Meanwhile the proportion of funds going to solicited projects has steadily increased from 20% in FY1998 to 50% in FY2021 (*Figure 6*, panel B).

Box-plot distributions over time of log-transformed costs of unsolicited (*Appendix 1—figure 4*, *panel A*) and solicited (*Appendix 1—figure 4*, *panel B*) projects show variations in whisker ranges

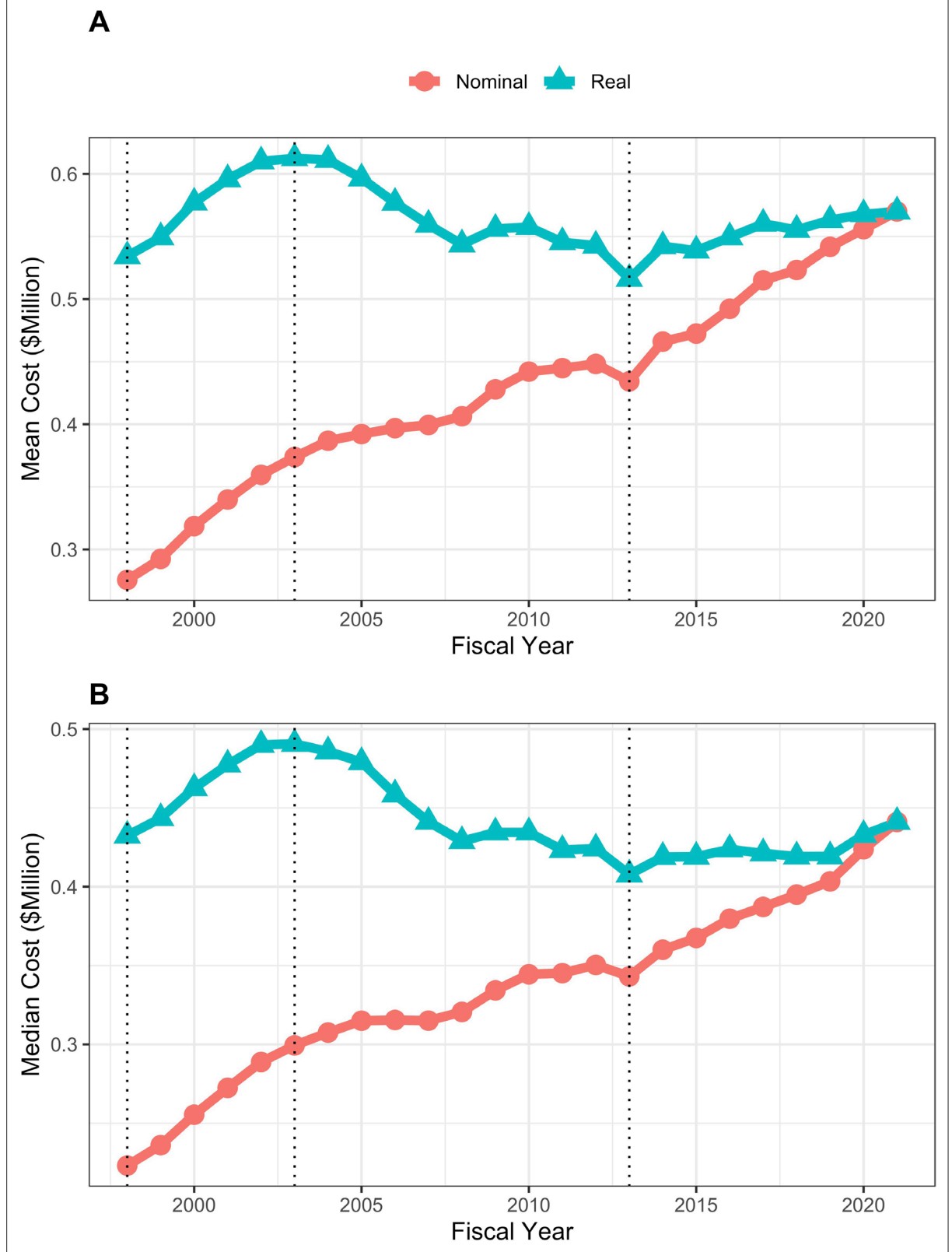

**Figure 2.** Mean (panel A) and median (panel B) nominal and real (BRDPI-adjusted in 2021 dollars) costs for NIH-funded RPGs, FY1998 to FY2021. The vertical dotted lines refer to the beginning and end of the NIH budget-doubling (FY1998 and FY2003) and the year of budget sequestration (FY2013).

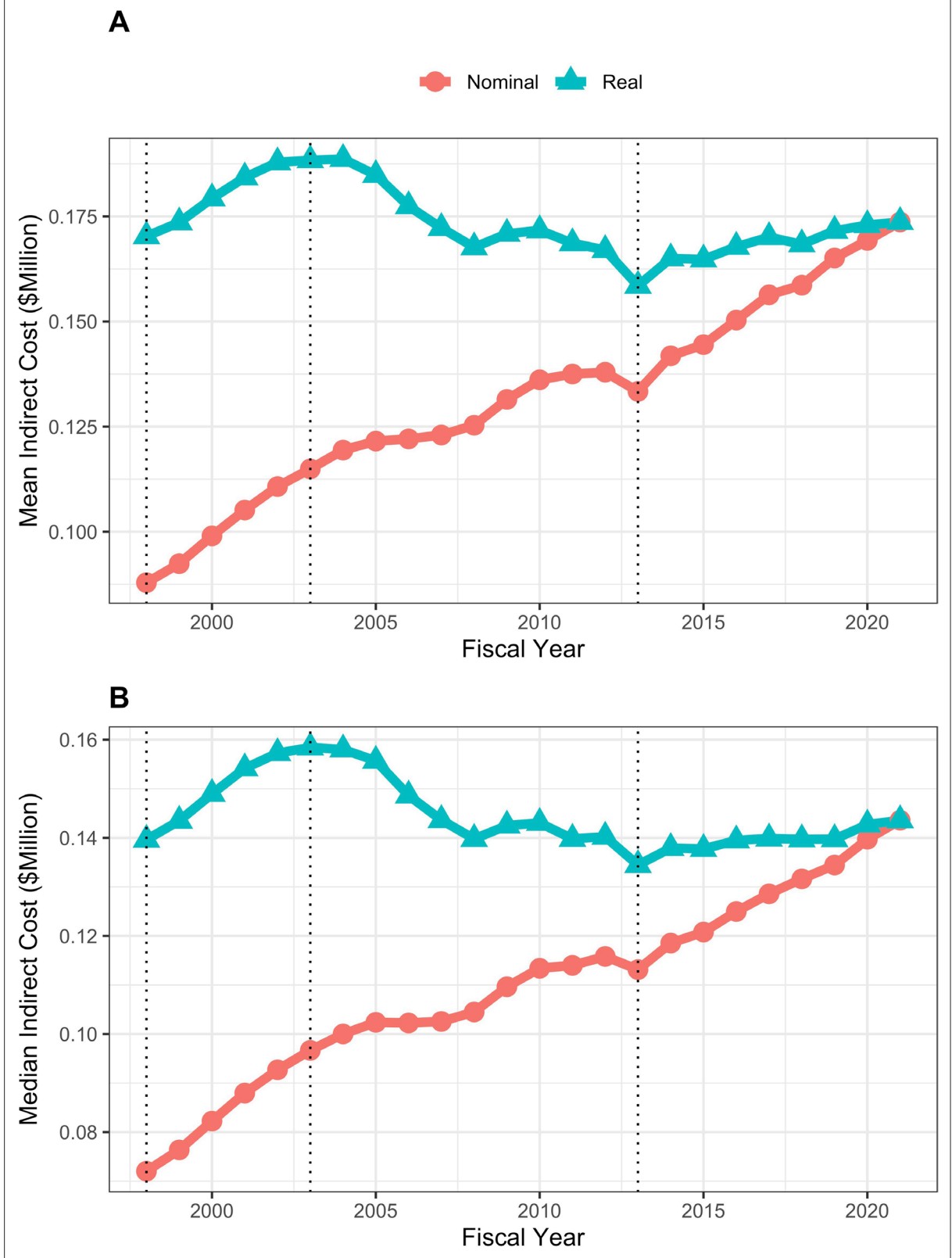

**Figure 3.** Mean (panel A) and median (panel B) nominal and real (BRDPI-adjusted in 2021 dollars) indirect costs for NIH-funded RPGs, FY1998 to FY2021. TThe vertical dotted lines refer to the beginning and end of the NIH budget-doubling (FY1998 and FY2003) and the year of budget sequestration (FY2013).

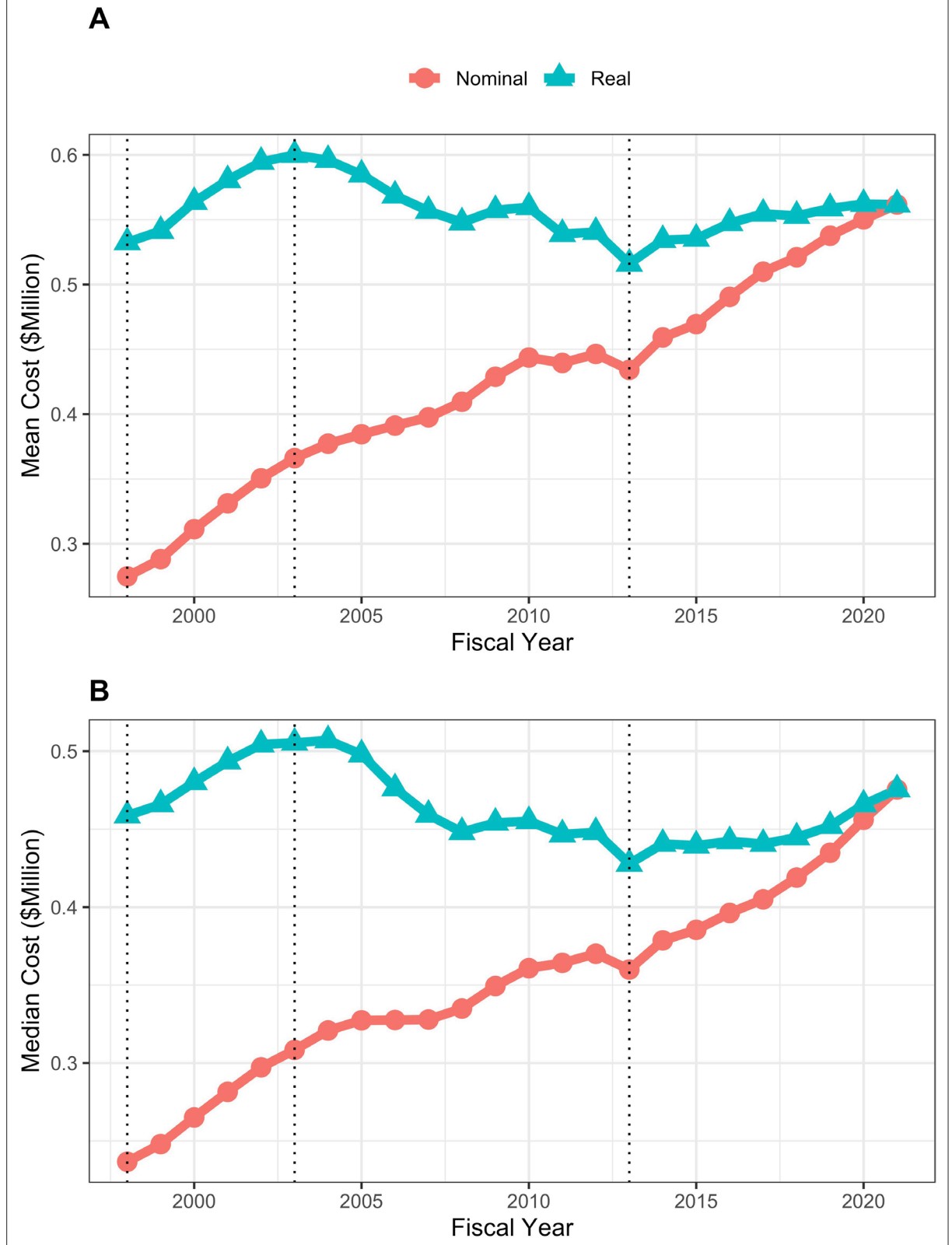

**Figure 4.** Mean (panel A) and median (panel B) nominal and real (BRDPI-adjusted in 2021 dollars) costs for NIH-funded R01-equivalent awards, FY1998 to FY2021. The vertical dotted lines refer to the beginning and end of the NIH budget-doubling (FY1998 and FY2003) and the year of budget sequestration (FY2013).

**Table 1.** Characteristics of RPGs in selected fiscal years.

| Characteristic | | 1998 | 2003 | 2013 | 2019 | 2021 |
|---|---|---|---|---|---|---|
| Total N (%) | | 26882 (15.5) | 35513 (20.5) | 33047 (19.1) | 38241 (22.1) | 39513 (22.8) |
| Total Costs ($M BRDPI) | Mean (SD) | 0.53 (0.58) | 0.61 (0.76) | 0.52 (0.70) | 0.56 (0.88) | 0.57 (0.90) |
| Log-10 Total Costs BRDPI | Mean (SD) | 5.63 (0.27) | 5.69 (0.26) | 5.62 (0.25) | 5.64 (0.27) | 5.65 (0.26) |
| Indirect Costs ($M BRDPI) | Mean (SD) | 0.17 (0.17) | 0.19 (0.19) | 0.16 (0.16) | 0.17 (0.22) | 0.17 (0.21) |
| Unsolicited | Yes | 21905 (81.5) | 25913 (73.0) | 23300 (70.5) | 24153 (63.2) | 23471 (59.4) |
| R01-Equivalent | Yes | 21543 (80.1) | 30124 (84.8) | 26199 (79.3) | 29341 (76.7) | 31258 (79.1) |
| R21 or R03 | Yes | 1424 (5.3) | 3920 (11.0) | 4805 (14.5) | 5478 (14.3) | 4949 (12.5) |
| Program Grant | Yes | 811 (3.0) | 984 (2.8) | 595 (1.8) | 374 (1.0) | 328 (0.8) |
| Cooperative Agreement | Yes | 623 (2.3) | 1493 (4.2) | 1709 (5.2) | 2437 (6.4) | 2404 (6.1) |
| Costs > $5 million BRDPI | Yes | 38 (0.1) | 102 (0.3) | 70 (0.2) | 140 (0.4) | 135 (0.3) |
| Clinical Trial | Yes | | | 2389 (7.2) | 3706 (9.7) | 4176 (10.6) |
| Human or Animal | Neither | 5715 (21.3) | 6799 (19.1) | 6197 (18.8) | 7207 (18.8) | 7627 (19.3) |
| | Animal | 11357 (42.2) | 14877 (41.9) | 14842 (44.9) | 16231 (42.4) | 16633 (42.1) |
| | Human | 7063 (26.3) | 10529 (29.6) | 9424 (28.5) | 11275 (29.5) | 11820 (29.9) |
| | Both | 2747 (10.2) | 3308 (9.3) | 2584 (7.8) | 3528 (9.2) | 3433 (8.7) |
| Organization Type | Institute of Higher Education | 22005 (81.9) | 28911 (81.4) | 27041 (81.8) | 31434 (82.2) | 32564 (82.4) |
| | Research Organization | 2287 (8.5) | 3105 (8.7) | 2626 (7.9) | 2634 (6.9) | 2598 (6.6) |
| | Independent Hospital | 2097 (7.8) | 2634 (7.4) | 2641 (8.0) | 3367 (8.8) | 3495 (8.8) |
| | Other | 493 (1.8) | 863 (2.4) | 739 (2.2) | 806 (2.1) | 856 (2.2) |

RPG = Research Project Grant. BRDPI = Biomedical Research and Development Price Index. All BRDPI-adjusted costs are based on an FY2021 reference (in other words, based on 2021 dollars).

(*Figures 7 and 8*), but throughout time solicited projects have much greater degrees of variation as reflected in larger whisker ranges (*Figure 8*); in more recent years the whisker ranges for solicited projected, while still much higher than for unsolicited projects, have decreased (*Figure 8*).

We compared solicited and unsolicited projects in FY2021 and FY2010 (*Table 3*). In FY2021 solicited projects were more expensive (mean of $710,000 versus $480,000), and more likely to be over $5 million, to be a cooperative agreement, to be a clinical trial, and to involve human participants. Solicited projects were also more likely to be funded through small R21 or R03 mechanisms, while much less likely to be funded via an R01-equivalent mechanism. Thus, the wide whisker ranges of solicited projects (*Figure 8*), which have become more common over time (*Figure 6*, panel B), may reflect both expensive and inexpensive awards. Inexpensive R21 and R03 awards have increased from 5% of projects in FY1998 to nearly 16% in FY2015, with a modest decline since (*Appendix 1—figure 5*).

We similarly compared solicited and unsolicited R01-equivalent awards in FY2021 and FY2010 (*Table 4*). Solicited R01-equivalent awards were more expensive and more likely to involve clinical trials and human participants. Consistent with higher costs, they were less likely to have nominal direct costs less than $250,000 and more likely to have nominal direct costs greater than $500,000.

## Other RPG characteristics and costs over time

RPGs involving clinical trials are more expensive but, at least, over the last 10 years real costs remain stable (*Appendix 1—figure 6*). We acknowledge, though, that these analyses do not consider trial types, designs, or measures like numbers of patients enrolled. Real-cost trends among RPGs are similar

**Table 2.** Characteristics of R01 equivalent grants in selected fiscal years.

| Characteristic | | 1998 | 2003 | 2013 | 2019 | 2021 |
|---|---|---|---|---|---|---|
| Total N (%) | | 21543 (15.6) | 30124 (21.8) | 26199 (18.9) | 29341 (21.2) | 31258 (22.6) |
| Total Costs ($M BRDPI) | Mean (SD) | 0.53 (0.48) | 0.60 (0.70) | 0.52 (0.48) | 0.56 (0.52) | 0.56 (0.50) |
| Log-10 Total Costs BRDPI | Mean (SD) | 5.67 (0.19) | 5.72 (0.20) | 5.66 (0.18) | 5.69 (0.19) | 5.70 (0.19) |
| Nominal Direct Costs ($M) | Mean (SD) | 0.19 (0.20) | 0.25 (0.36) | 0.30 (0.35) | 0.37 (0.41) | 0.39 (0.41) |
| Nominal Direct Costs $250 K or Less | Yes | 18962 (88.0) | 23701 (78.7) | 16288 (62.2) | 11824 (40.3) | 10351 (33.1) |
| Nominal Direct Costs $500 K or More | Yes | 485 (2.3) | 1243 (4.1) | 1652 (6.3) | 3625 (12.4) | 4450 (14.2) |
| Indirect Costs ($M BRDPI) | Mean (SD) | 0.17 (0.12) | 0.19 (0.15) | 0.16 (0.10) | 0.17 (0.13) | 0.18 (0.12) |
| Unsolicited | Yes | 18445 (85.6) | 24337 (80.8) | 19873 (75.9) | 20642 (70.4) | 20637 (66.0) |
| Costs > $5 million BRDPI | Yes | 23 (0.1) | 69 (0.2) | 33 (0.1) | 41 (0.1) | 47 (0.2) |
| Clinical Trial | Yes | | | 1818 (6.9) | 2581 (8.8) | 3046 (9.7) |
| Human or Animal | Neither | 4612 (21.4) | 5923 (19.7) | 4830 (18.4) | 5299 (18.1) | 5773 (18.5) |
| | Animal | 9331 (43.3) | 12966 (43.0) | 12209 (46.6) | 12991 (44.3) | 13621 (43.6) |
| | Human | 5535 (25.7) | 8632 (28.7) | 7110 (27.1) | 8207 (28.0) | 9023 (28.9) |
| | Both | 2065 (9.6) | 2603 (8.6) | 2050 (7.8) | 2844 (9.7) | 2841 (9.1) |
| Organization Type | Institute of Higher Education | 17646 (81.9) | 24616 (81.7) | 21461 (81.9) | 24119 (82.2) | 25811 (82.6) |
| | Research Organization | 1814 (8.4) | 2567 (8.5) | 2068 (7.9) | 2006 (6.8) | 2007 (6.4) |
| | Independent Hospital | 1679 (7.8) | 2218 (7.4) | 2076 (7.9) | 2597 (8.9) | 2784 (8.9) |
| | Other | 404 (1.9) | 723 (2.4) | 594 (2.3) | 619 (2.1) | 656 (2.1) |

BRDPI = Biomedical Research and Development Price Index. All BRDPI-adjusted costs are based on an FY2021 reference (in other words, based on 2021 dollars).

irrespective of human or animal classification, though as expected projects involving human participants or human participants and animal models were more expensive than others (*Appendix 1—figure 7*).

## Independent association of time with BRDPI-adjusted RPG costs

We conducted a series of regression analyses to examine whether there may be an association of time (that is fiscal year) with BRDPI-adjusted costs of RPG projects separate from those associated with funding mechanism, solicitation (or not), involvement of human participant or animal models, or type of recipient organization. We attempted multivariable linear regressions with log-10 transformed costs as the dependent variable (*Appendix 1—table 1*; *Leifeld, 2013*), but upon inspection of residual diagnostics found poor model fit due to fat-tailed distributions. By fat-tailed we mean that many values were far from the mean or median without being outliers; one can think of a 'bell-shaped' curve that is substantively widened. We looked into other possible transformations (e.g. arcsinh, Box-Cox, center and scale, exponential, square-root, and Yeo-Johnson) and did not find substantive improvements. We therefore performed a wholly non-parametric random forest regression (*Ishwaran and Kogalur, 2022*) of log-10 transformed total costs. By 'non-parametric' we mean that there are no pre-specified patterns such as a linear relationship between costs and putative explanatory variables. The random forest method is one type of machine learning that allows for extensive validation and for interactions between variables. (*Breiman, 2001*) The model, based on a one-percent random sample, performed well, able to explain over 47% of the variance of costs. Time (that is fiscal year) contributed little to prediction (*Appendix 1—table 2*). *Figure 9* overlays the multivariable adjusted per-project total costs with actual observed median total costs and shows no material difference.

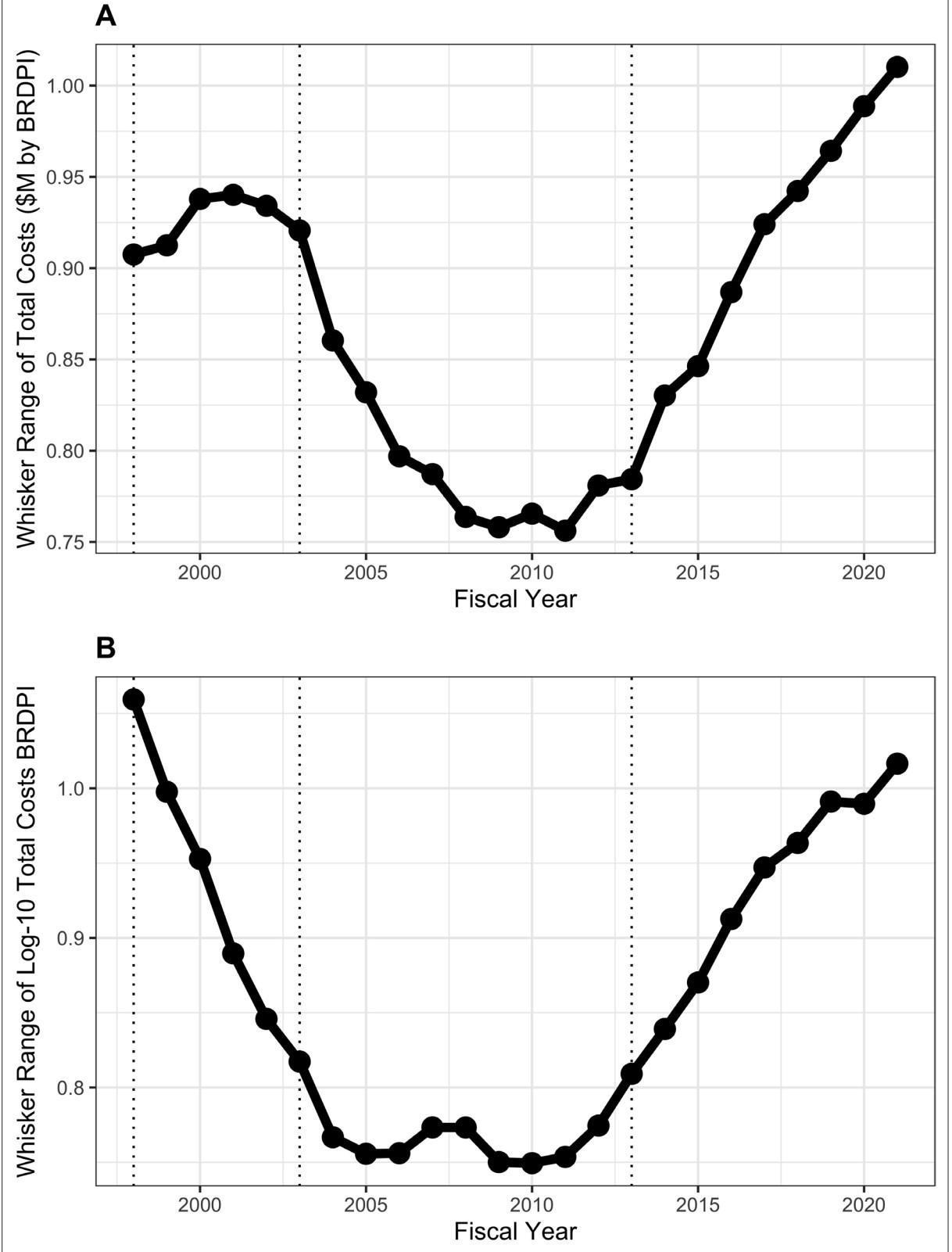

**Figure 5.** Whisker ranges, that is the distance between the tips of upper and lower whiskers, of box plots shown in *Appendix 1—figure 2*. All costs are FY2021 BRDPI-adjusted. Panel A is based on untransformed real costs, while Panel B is based on log-10 transformed real costs. The vertical dotted lines refer to the beginning and end of the NIH budget-doubling (FY1998 and FY2003) and the year of budget sequestration (FY2013).

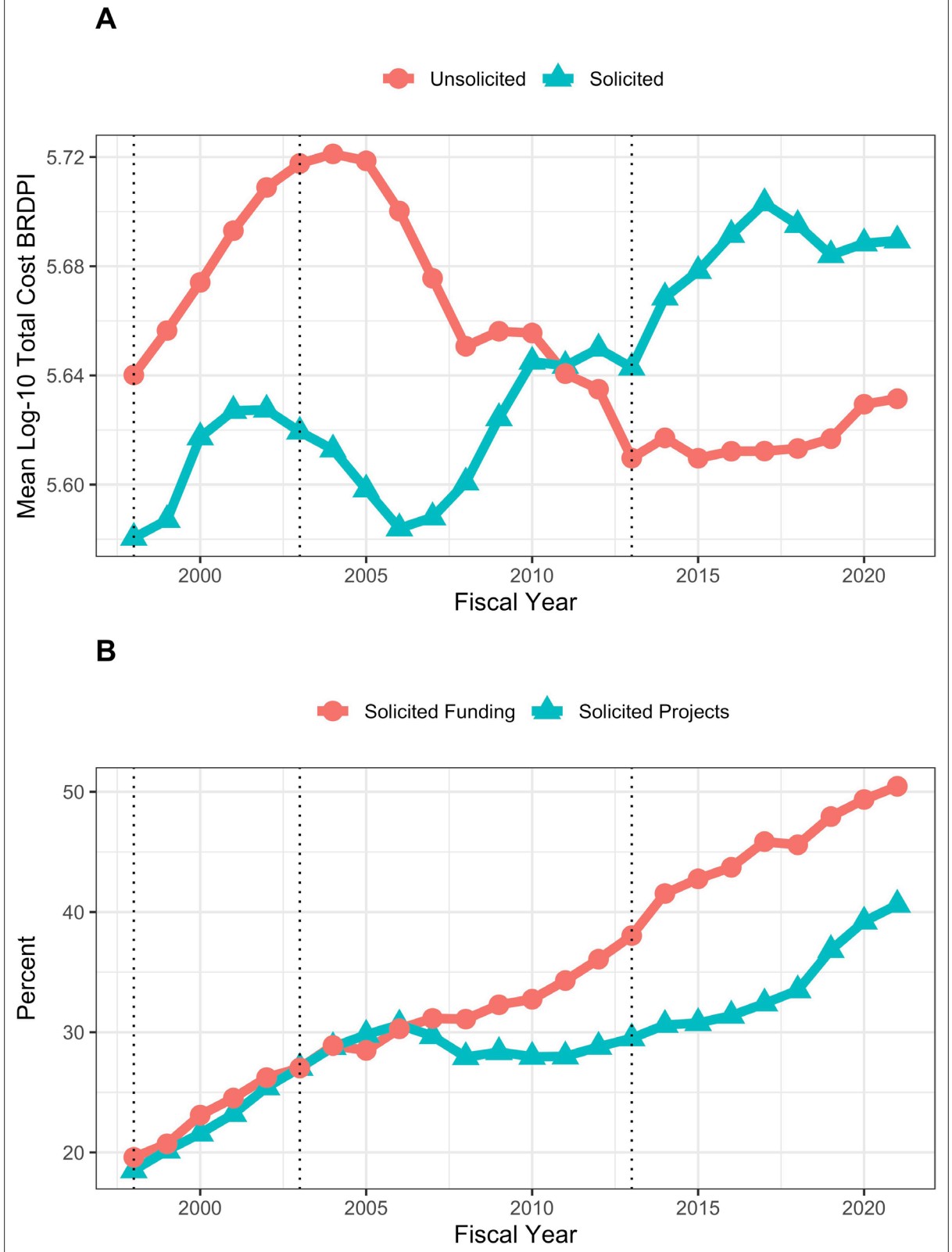

**Figure 6.** Trends in solicited and unsolicited RPGs FY1998 to FY2021. (**A**): Log-transformed costs according to NIH solicitation. All costs are FY2021 BRDPI-adjusted. (**B**): Percent of RPG projects and percent of RPG funding going to solicited awards. The vertical dotted lines refer to the beginning and end of the NIH budget-doubling (FY1998 and FY2003) and the year of budget sequestration (FY2013).

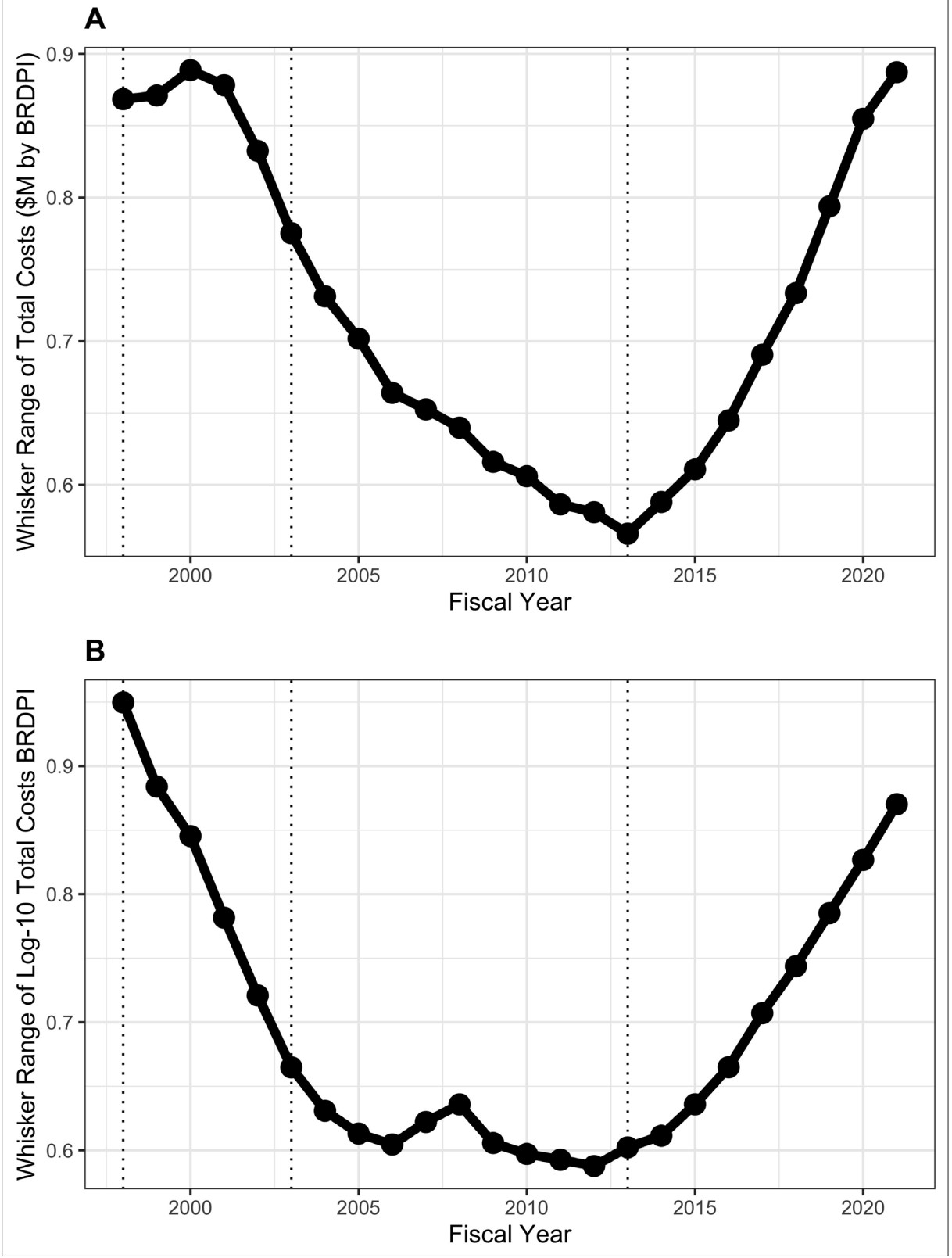

**Figure 7.** Whisker ranges, that is the distance between the tips of upper and lower whiskers of box plots showing distribution of costs of unsolicited RPG awards (*Appendix 1—figure 4*). All costs are FY2021 BRDPI-adjusted. Panel A is based on untransformed real costs, while Panel B is based on log-10 transformed real costs. The vertical dotted lines refer to the beginning and end of the NIH budget-doubling (FY1998 and FY2003) and the year of budget sequestration (FY2013).

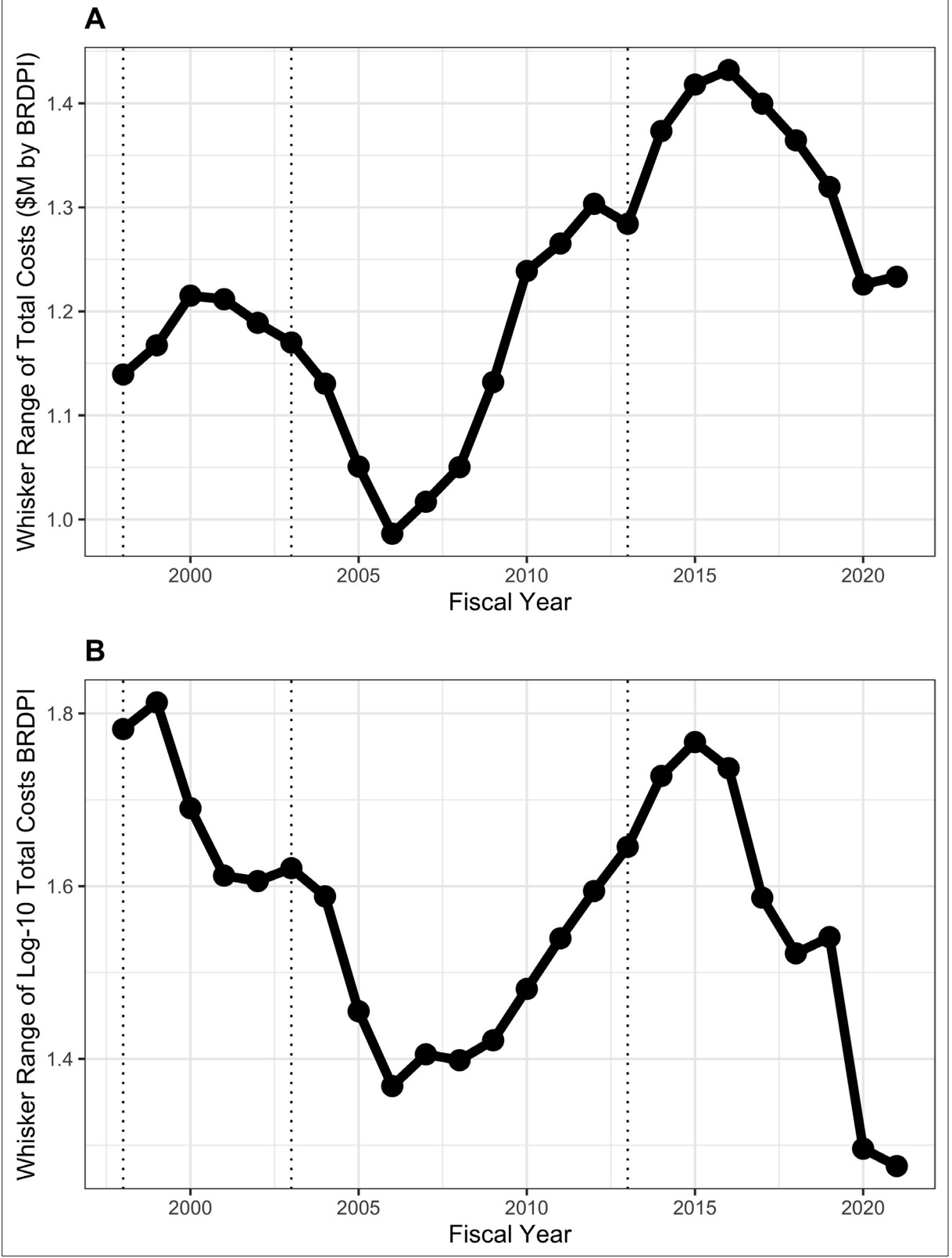

**Figure 8.** Whisker ranges, that is the distance between the tips of upper and lower whiskers of box plots showing distribution of costs of solicited RPG awards (*Appendix 1—figure 4*). All costs are FY2021 BRDPI-adjusted. Panel A is based on untransformed real costs, while Panel B is based on log-10 transformed real costs. The vertical dotted lines refer to the beginning and end of the NIH budget-doubling (FY1998 and FY2003) and the year of budget sequestration (FY2013).

**Table 3.** Characteristics of FY2021 and FY2010 RPGs according to solicitation status.

| Characteristic | | Solicited 2021 | Unsolicited 2021 | Solicited 2010 | Unsolicited 2010 |
|---|---|---|---|---|---|
| Total N (%) | | 16042 (40.6) | 23471 (59.4) | 9697 (28.0) | 24986 (72.0) |
| Total Costs ($M BRDPI) | Mean (SD) | 0.71 (1.36) | 0.48 (0.31) | 0.65 (1.28) | 0.52 (0.47) |
| Log-10 Total Costs BRDPI | Mean (SD) | 5.69 (0.33) | 5.63 (0.20) | 5.65 (0.35) | 5.66 (0.20) |
| R01-Equivalent | Yes | 10621 (66.2) | 20637 (87.9) | 6677 (68.9) | 22196 (88.8) |
| R21 or R03 | Yes | 2750 (17.1) | 2199 (9.4) | 2248 (23.2) | 1857 (7.4) |
| Costs > $5 million BRDPI | Yes | 121 (0.8) | 14 (0.1) | 55 (0.6) | 32 (0.1) |
| Cooperative Agreement | Yes | 2327 (14.5) | 77 (0.3) | 1478 (15.2) | 208 (0.8) |
| Clinical Trial | Yes | 3104 (19.3) | 1072 (4.6) | 1335 (13.8) | 1067 (4.3) |
| Human or Animal | Neither | 3407 (21.2) | 4220 (18.0) | 1632 (16.8) | 5096 (20.4) |
| | Animal | 4579 (28.5) | 12054 (51.4) | 3151 (32.5) | 12696 (50.8) |
| | Human | 6823 (42.5) | 4997 (21.3) | 4287 (44.2) | 5347 (21.4) |
| | Both | 1233 (7.7) | 2200 (9.4) | 627 (6.5) | 1847 (7.4) |
| Organization Type | Institute of Higher Education | 12983 (80.9) | 19581 (83.4) | 7662 (79.0) | 20848 (83.4) |
| | Research Organization | 1214 (7.6) | 1384 (5.9) | 902 (9.3) | 1836 (7.3) |
| | Independent Hospital | 1419 (8.8) | 2076 (8.8) | 824 (8.5) | 1824 (7.3) |
| | Other | 426 (2.7) | 430 (1.8) | 309 (3.2) | 478 (1.9) |

RPG = Research Project Grant. BRDPI = Biomedical Research and Development Price Index. All BRDPI-adjusted costs are based on an FY2021 reference (in other words, basedon 2021 dollars).

## Discussion

The rate of inflation for NIH-funded research (that is the BRDPI) was higher than the general rate of inflation from FY1998 until FY2012; since then, the rate of inflation for NIH-funded research has been similar to the general rate of inflation. The BRDPI is determined via a sophisticated methodology; since 2005 the Bureau of Economic Analysis (BEA) uses a Fisher chain-weighted indexed methodology which is analogous to calculating compound growth on retirement portfolios over many years as the mix of stocks and bonds changes from year to year. The decrease in the BRDPI in FY2012 was likely related to an NIH-imposed salary cap 'freeze' in 2011. In 2012, the NIH has linked the salary cap to Executive Level II (instead of the higher Executive Level I) salaries. Since then, salary caps continue to linked to Executive Level II levels and have increased at the rate of Federal civilian salaries, which likely have risen a rate lower than academic salaries. The cap reductions in FY2011, the relatively slow rate of rise of Federal salaries which determine the NIH salary cap, along with relatively low increases in fellowship and training stipends have combined to reduce the BRDPI since FY2011 (**NIH, 2022c**). Institutions and faculty may be under greater pressures as the differential between NIH-imposed salary caps and actual faculty salaries increases (**NIH, 2022c**). They are also facing pressures due to increasing competition for post-doctoral research fellows who realize greater shorter and longer term economic success outside of the academy (**Kahn and Ginther, 2017**).

Real (BRDPI-adjusted) average and median RPG costs increased during the NIH-doubling (FY1998 to FY2003), decreased after the doubling and have remained relatively stable since. Of note, though, the degree of variation of RPG costs has changed over time, with more marked extremes observed on both higher and lower levels of cost. On the higher end, over time NIH has been funding more cooperative agreements, more projects exceeding $5 million (in FY2021 BRDPI, *not nominal*, values), and more clinical trials. The top centile of projects are receiving a substantially greater share of the overall RPG funding pool. On the lower end of cost, over time the agency has been funding more low-cost mechanism awards (R03 and R21). On both ends of the cost spectrum, the agency is funding a greater proportion of solicited projects, with nearly half of RPG money going towards solicited projects. These compositional changes likely reflect evolving priorities articulated in NIH strategic

**Table 4.** Characteristics of FY2021 and FY2010 R01 equivalent grant awards according to solicitation status.

| Characteristic | | Solicited 2021 | Unsolicited 2021 | Solicited 2010 | Unsolicited 2010 |
|---|---|---|---|---|---|
| Total N (%) | | 10621 (34.0) | 20637 (66.0) | 6677 (23.1) | 22,196 (76.9) |
| Total Costs ($M BRDPI) | Mean (SD) | 0.67 (0.73) | 0.51 (0.30) | 0.74 (1.19) | 0.50 (0.38) |
| Log-10 Total Costs BRDPI | Mean (SD) | 5.74 (0.24) | 5.67 (0.15) | 5.75 (0.27) | 5.67 (0.15) |
| Nominal Direct Costs ($M) | Mean (SD) | 0.47 (0.62) | 0.34 (0.23) | 0.42 (0.78) | 0.27 (0.26) |
| Nominal Direct Costs $250 K or Less | Yes | 3049 (28.7) | 7302 (35.4) | 3121 (46.7) | 15,702 (70.7) |
| Nominal Direct Costs $500 K or More | Yes | 2460 (23.2) | 1990 (9.6) | 1191 (17.8) | 832 (3.7) |
| Costs > $5 million BRDPI | Yes | 35 (0.3) | 12 (0.1) | 44 (0.7) | 26 (0.1) |
| Clinical Trial | Yes | 2126 (20.0) | 920 (4.5) | 976 (14.6) | 927 (4.2) |
| Human or Animal | Neither | 2337 (22.0) | 3436 (16.6) | 1017 (15.2) | 4525 (20.4) |
| | Animal | 2902 (27.3) | 10719 (51.9) | 2202 (33.0) | 11,451 (51.6) |
| | Human | 4656 (43.8) | 4367 (21.2) | 3055 (45.8) | 4686 (21.1) |
| | Both | 726 (6.8) | 2115 (10.2) | 403 (6.0) | 1534 (6.9) |
| Organization Type | Institute of Higher Education | 8650 (81.4) | 17161 (83.2) | 5259 (78.8) | 18,519 (83.4) |
| | Research Organization | 791 (7.4) | 1216 (5.9) | 632 (9.5) | 1639 (7.4) |
| | Independent Hospital | 904 (8.5) | 1880 (9.1) | 579 (8.7) | 1604 (7.2) |
| | Other | 276 (2.6) | 380 (1.8) | 207 (3.1) | 434 (2.0) |

BRDPI = BiomedicalResearch and Development Price Index. All BRDPI-adjusted costs are based on an FY2021 reference (in other words, based on 2021dollars).

planning documents.(**NIH, 2023b**) Despite increases in nominal costs and despite increased proportions of funding going to solicited projects, recent years have seen increases in the absolute numbers of RPG and R01 awards. After adjusting for potential confounders in a wholly non-parametric machine learning regression, we find no independent association of time with BRDPI-adjusted costs. Recalling the automobile rental firm analogy, NIH may be pursuing the strategy of simultaneously purchasing more expensive (large vans) and less expensive (compact cars) vehicles, reflective of changing priorities and compositional effects over time.

## Why are costs for services (and research) so high?

Increases in costs for research may be greater than increases in general economy-wide costs just as educational and health-care costs have increased at rates much greater than other costs. The Nobel-prize winning economist William Baumol explored differential increases in costs in his work on 'the cost disease'. (**Malach and Baumol, 2012**) The fundamental problem is that different sectors of the economy realize different rates of improvements in productivity. Baumol cites 4 musicians who play a Beethoven string quartet; there has been no change in productivity between 1826 and now. It takes just as many musicians just as much time to 'produce' a live performance of a Beethoven string quartet. However, in other segments of the economy, productivity has increased dramatically, leading to increased wages for non-string-quartet workers. If we still want live performances of string quartets

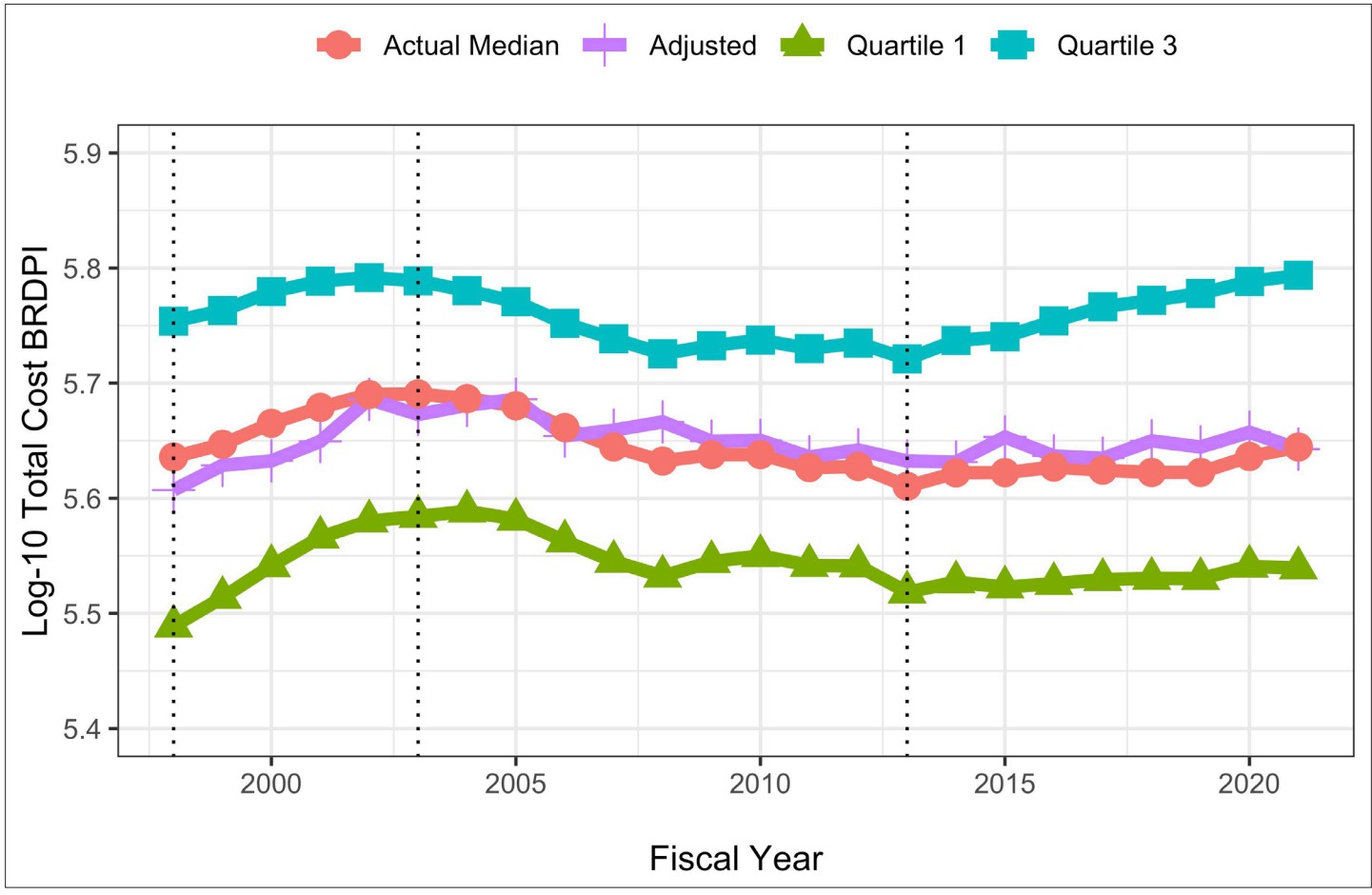

**Figure 9.** Actual and random-forest regression log-transformed costs for NIH-funded RPGs, FY1998 to FY2021. All costs are FY2021 BRDPI-adjusted. The vertical dotted lines refer to the beginning and end of the NIH budget-doubling (FY1998 and FY2003) and the year of budget sequestration (FY2013).

we have to pay much more now than in 1826 even though the output is unchanged because otherwise the musicians will choose other lines of work that pay more. The economists Eric Helland (Claremont McKenna College, RAND) and Alex Tabarrok (George Mason University) posted a report entitled 'Why are the Prices so D-mn High?' in which they explain how Baumol's construct works for explaining cost increases in the service sector, and in education and healthcare in particular (**Helland and Tabarrok, 2019**). It is important to note, though, that the NIH caps on salaries and salary increases since 2012 may well have mitigated the effects of the cost disease on the NIH portfolio.

Helland and Tabarrok illustrate the problem (**Helland and Tabarrok, 2019**) by imagining a simple two-product economy that produces only one good – cars – and one service – education. If society wants more education, the the opportunity cost (or price) will be fewer cars. Over time, productivity improves for both cars and education, but to a much greater degree for cars. If society wants to maintain the same ratio of education to cars, the price for that education relative to cars will be much higher. If society wants more education the price for education will be higher still. Thus, over time, relative prices for services (education) increase while prices for goods (cars) decline.

Bureau of Economic Analysis data (**Helland and Tabarrok, 2019**) on the relative costs of goods and services in the United States since 1950 show that the United States economy has shifted from goods to services while the relative prices for services (like education and healthcare) have increased. There is literature on the costs and productivity of research showing similar long-term patterns. For example, Scannell et al described 'Eroom's Law' of *declining efficiency* of pharmaceutical research and development dating back to 1950 and continuing relentlessly since. (**Scannell et al., 2012**) The number of drugs developed per billion dollars of R&D spending has *declined* by at least an order of magnitude.

Other recent work has focused on the increasing costs of conducting clinical trials, (*Sertkaya et al., 2016*) whether sponsored by industry or by NIH. (*Lauer et al., 2017*) This literature identifies other drivers specific to pharmaceutical research or clinical trials in general, but these drivers may reflect general longstanding and inherent increases in the prices of services.

### Limitations

While we are able to describe changes in RPG costs over time, we note a number of important limitations. Our analyses are based on NIH as an agency; each Institute and Center has its own strategic plans and priorities. There is not a simple one-to-one link between specific grants and projects. Some projects are supported by multiple sources, including some outside of NIH. Individual grants are only partially able to cover costs, especially indirect costs for which recovery is nearly always partial. Because of salary caps and heterogeneous practices by which institutions use NIH funds for salary support, we do not have comprehensive information on compensation for personnel. Other investigators have leveraged university-based data to document how federal funds are used to directly compensate researchers and to enable researchers not directly supported on grants to publish their work. (*Sattari et al., 2022*) Our regression analyses could only account for those variables we have in hand; nonetheless, the random forest model was able to account for a substantial proportion of the variance in RPG costs. While our analyses demonstrate that a greater proportion of funds is going to large-scale solicited projects, further work will be needed to determine whether this shift is translating into greater productivity or scientific advances.

## Materials and methods

BRDPI and GDP-index values were obtained from the NIH Office of the Budget (*NIH, 2022c*). We queried Research Project Grant (RPG) data from NIH IMPAC II files. RPGs were defined as those grants with activity codes of DP1, DP2, DP3, DP4, DP5, P01, PN1, PM1, R00, R01, R03, R15, R21, R22, R23, R29, R33, R34, R35, R36, R37, R61, R50, R55, R56, RC1, RC2, RC3, RC4, RF1, RL1, RL2, RL9, RM1, SI2, UA5, UC1, UC2, UC3, UC4, UC7, UF1, UG3, UH2, UH3, UH5, UM1, UM2, U01, U19, U34 and U3R. Not all of these activity codes were used by NIH every year. R01-equivalent awards were defined as activity codes DP1, DP2, DP5, R01, R37, R56, RF1, RL1, U01 and R35 from select NIGMS and NHGRI program announcements. Not all of these activities may be in use by NIH every year. For FY2009 and FY2010 we excluded awards made under the American Recovery and Reinvestment Act of 2009 (ARRA) and all ARRA solicited applications and awards. For FY2020 and FY2021 we excluded awards issued using supplemental Coronavirus (COVID-19) appropriations.

## Additional information

### Funding
No external funding was received for this work.

### Author contributions
Michael S Lauer, Conceptualization, Formal analysis, Supervision, Writing – original draft; Joy Wang, Deepshikha Roychowdhury, Formal analysis, Writing - review and editing

### Author ORCIDs
Michael S Lauer http://orcid.org/0000-0002-9217-8177

### Decision letter and Author response
Decision letter https://doi.org/10.7554/eLife.84245.sa1
Author response https://doi.org/10.7554/eLife.84245.sa2

## Additional files

### Supplementary files

• MDAR checklist

• Source data 1. BRDPI_change_2021 xlsx; data file with values for the Biomedical Research and Development Price Index (BRDPI) used to generate *Figure 1*, panel B.

• Source data 2. GDP_change_2021 xlsx; data file with value for the Gross Domestic Product (GDP) Price Index used to generate *Figure 1*, panel B.

• Source data 3. rpg_anon_id RData; anonymized source data (in *R* format) used to generate all text, tables, and figures (except *Figure 1*, panel B and the red line with circles in Figure 1, panel A) in the main manuscript and appendix.

• Source data 4. awardee_summary RData; source data (in *R* format) used to generate the red line (with circles) in *Figure 1*, panel A.

• Source code 1. RPG inflation 1 16 23 Rmd: *R* markdown file which contains a data dictionary and code used to generate all text, tables, and figures in the main manuscript.

• Source code 2. RPG inflation 1 16 23 appendix Rmd: *R* markdown file which contains a data dictionary and code used to generate all text, tables, and figures in the appendix.

### Data availability

Anonymized source data (in Excel and .RData formats) have been provided as supplementary files. R markdown source code for the main paper and the appendix corresponds with all numbers, tables, and figures. There are no restrictions to use.

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

## Appendix 1

### Research Project Grant Cost and Compositional Changes Over Time: Descriptive Analyses

From 1998 through 2021, the absolute numbers of R01-equivalent awards followed a similar pattern as those of Research Project Grants (RPGs) in general, rising during the doubling, falling from the end of the doubling until FY2015, and rising since then (*Appendix 1—figure 1*).

While the central tendencies (means and medians) since FY2008 have remained relatively stable (noting the fall in FY2013, the year of budget sequestration), the non-outlier ranges, as visualized through box plots, have been substantially higher during the time of the NIH doubling (FY1998-FY2003) and during recent years (*Appendix 1—figure 2*). The proportion of funds going to the top centile (by total costs) of RPGs has increased since FY2013, but the proportion going to the top decile has only increased modestly (*Appendix 1—figure 3*). Most of the variability in total costs appears to be manifest in unsolicited projects (*Appendix 1—figure 4*). The proportion of projects and funding going to small mechanisms (R21 or R03 awards) increased during and after the doubling (FY1998-FY2003) but has decreased in recent years (*Appendix 1—figure 5*). There have been no marked changes in costs related to clinical trials (*Appendix 1—figure 6*), but these data should be interpreted with caution as large-scale trials tend to be funded through consortia that typically involve multiple grants, cooperative agreements, and/or contracts. The proportion of funding going to human studies has increased over time (*Appendix 1—figure 7*).

### Research Project Grant Costs Over Time: Regression Analyses

Linear regression analyses of BRDPI-adjusted log-10 transformed costs over time showed the time alone (i.e., fiscal years) explained less than 1% of total variance (*Appendix 1—table 1*, Model 1). A model that included compositional elements could explain 47% of total variance (*Appendix 1—table 1*, Model 2), but regression diagnostics, in particular QQ plots, were concerning given the fat-tailed distribution of costs even after logarithmic transformation. We therefore conducted random forest (a machine-learning method) regression; time (i.e., fiscal years) was a relatively unimportant predictor of BRDPI-adjusted log-10 transformed costs (*Appendix 1—table 2*)

**Appendix 1—table 1.** Linear Regression Models for log-10 transformed BRDPI-adjusted (or real) total costs of RPGs funded from FY1998 to FY2021.

Values shown for each variable are regression coefficients (standard errors). Model 1 considers time (fiscal year) only, while Model 2 factors in other variables. Model 1 explained only 0.6% of total variance, while Model 2 explained 47% of variance. Regression diagnostics indicated poor fit due largely to fat tails, which were present even after log-10 transformation. Table constructed using the R package texreg; see *Leifeld, 2013*, Journal of Statistical Software, 55(8), 1–24.

| | Model 1 | Model 2 |
|---|---|---|
| Intercept | 5.629 (0.002)*** | 5.477 (0.002)*** |
| FY1999 | 0.013 (0.002)*** | 0.011 (0.002)*** |
| FY2000 | 0.033 (0.002)*** | 0.028 (0.002)*** |
| FY2001 | 0.049 (0.002)*** | 0.045 (0.002)*** |
| FY2002 | 0.059 (0.002)*** | 0.059 (0.002)*** |
| FY2003 | 0.062 (0.002)*** | 0.067 (0.002)*** |
| FY2004 | 0.061 (0.002)*** | 0.071 (0.002)*** |
| FY2005 | 0.054 (0.002)*** | 0.066 (0.002)*** |
| FY2006 | 0.035 (0.002)*** | 0.048 (0.002)*** |
| FY2007 | 0.020 (0.002)*** | 0.037 (0.002)*** |

*Appendix 1—table 1 Continued on next page*

*Appendix 1—table 1 Continued*

| | Model 1 | Model 2 |
|---|---|---|
| FY2008 | 0.008 (0.002)*** | 0.028 (0.002)*** |
| FY2009 | 0.018 (0.002)*** | 0.033 (0.002)*** |
| FY2010 | 0.023 (0.002)*** | 0.036 (0.002)*** |
| FY2011 | 0.012 (0.002)*** | 0.047 (0.002)*** |
| FY2012 | 0.010 (0.002)*** | 0.049 (0.002)*** |
| FY2013 | −0.010 (0.002)*** | 0.031 (0.002)*** |
| FY2014 | 0.004 (0.002) | 0.045 (0.002)*** |
| FY2015 | 0.002 (0.002) | 0.046 (0.002)*** |
| FY2016 | 0.008 (0.002)*** | 0.053 (0.002)*** |
| FY2017 | 0.013 (0.002)*** | 0.054 (0.002)*** |
| FY2018 | 0.012 (0.002)*** | 0.052 (0.002)*** |
| FY2019 | 0.012 (0.002)*** | 0.055 (0.002)*** |
| FY2020 | 0.023 (0.002)*** | 0.063 (0.002)*** |
| FY2021 | 0.026 (0.002)*** | 0.061 (0.002)*** |
| Unsolicited before 2010 | | −0.003 (0.001)*** |
| Unsolicited after 2010 | | −0.026 (0.001)*** |
| R01 Equivalent | | 0.115 (0.001)*** |
| Clinical Trial | | 0.040 (0.001)*** |
| Human Only | | 0.030 (0.001)*** |
| Animal Only | | 0.080 (0.001)*** |
| Both Human and Animal | | 0.067 (0.001)*** |
| Research Organization | | 0.056 (0.001)*** |
| Indepdent Hospital | | 0.034 (0.001)*** |
| Other Organization | | 0.029 (0.001)*** |
| Cooperative Agreement | | 0.263 (0.001)*** |
| Program Grant | | 0.741 (0.002)*** |
| R21 or R03 | | −0.278 (0.001)*** |
| $R^2$ | 0.006 | 0.471 |
| Adj. $R^2$ | 0.006 | 0.471 |
| Num. obs. | 827815 | 827815 |

*** $p < 0.001$; ** $p < 0.01$; * $p < 0.05$.

*** p < 0.001; ** p < 0.01; * p < 0.05.

**Appendix 1—table 2.** Results of random forest regression for log-10 transformed BRDPI-adjusted (or real) total costs of a 1% random sample of RPGs funded from FY1998 to FY2021.
A model that consider fiscal year as the sole variable only explained 0.3% of total variance. A model

that included the 10 variables listed in the table explained 47% of the total variance. The variable importance reflects the deterioration in prediction error when the value for a variable is determined randomly, as opposed to its actual value. Variables reflecting compositional factors (like the type of award mechanism) were much more important than time (that is fiscal year) in predicting real total grant costs.

| Variable | Importance |
| --- | --- |
| Program Grant | 0.2615546 |
| R21 or R03 | 0.0990313 |
| Cooperative Agreement | 0.0954948 |
| R01 Equivalent | 0.0599555 |
| Fiscal Year | 0.0093220 |
| Organizational Type | 0.0091677 |
| Human | 0.0070634 |
| Unsolicited | 0.0037442 |
| Clinical Trial | 0.0034817 |
| Animal | 0.0030754 |

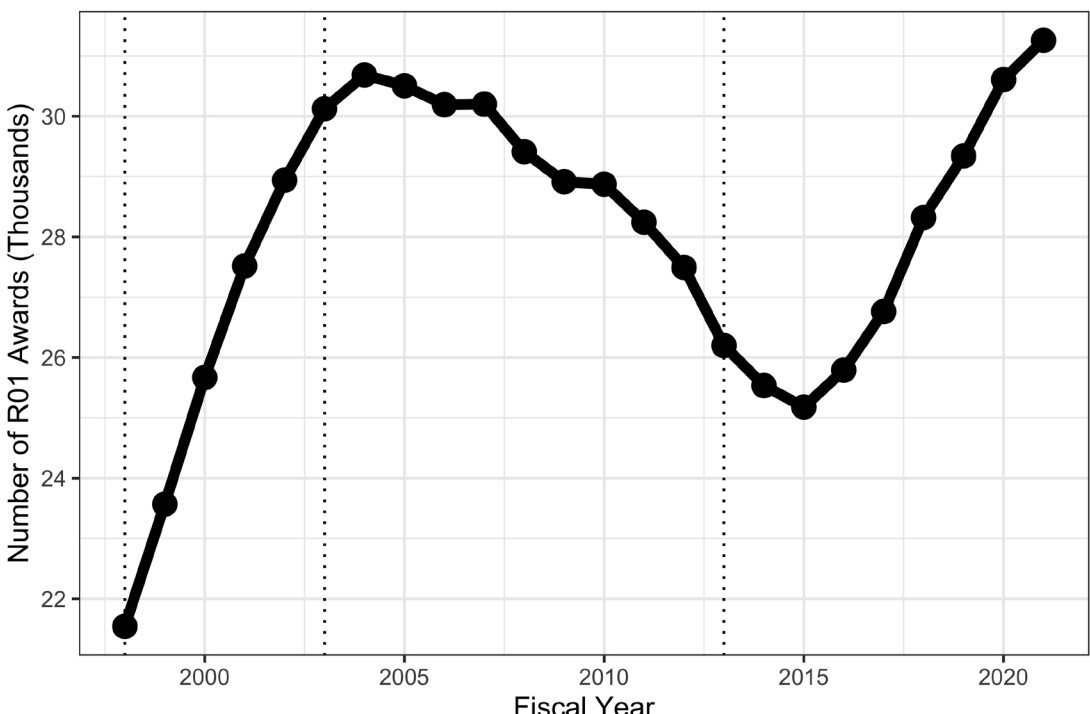

**Appendix 1—figure 1.** Number of funded R01-equivalent awards by fiscal year. The vertical dotted lines refer to the beginning and end of the NIH budget-doubling (FY1998 and FY2003) and the year of budget sequestration (FY2013).

A

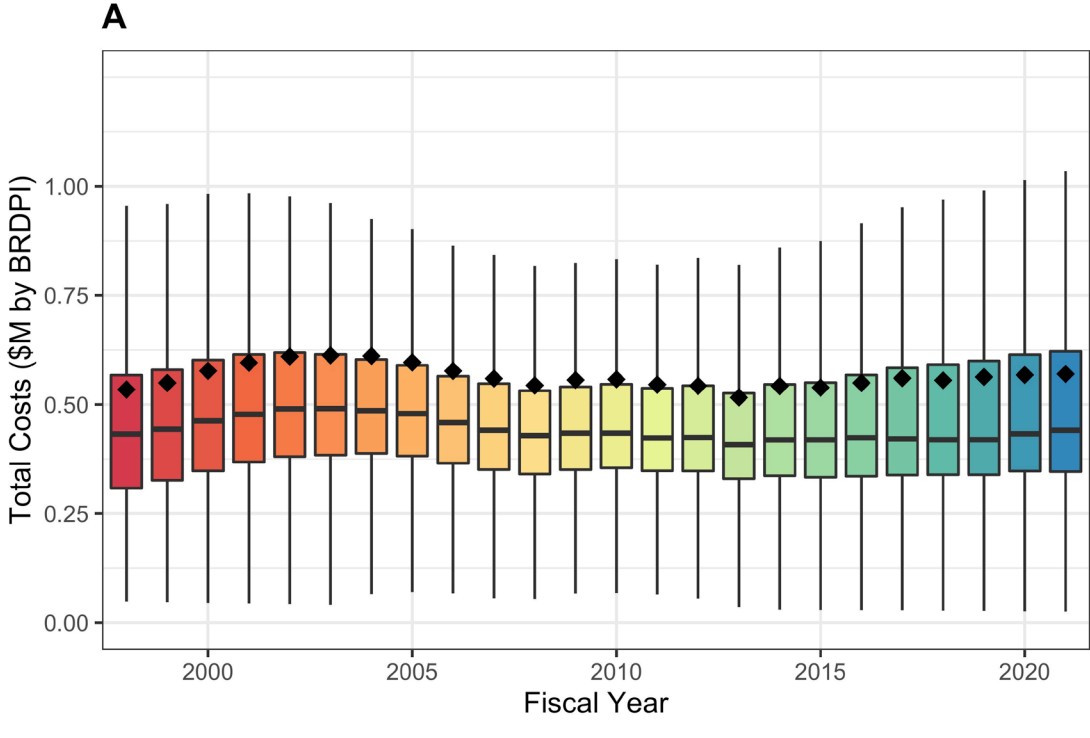

B

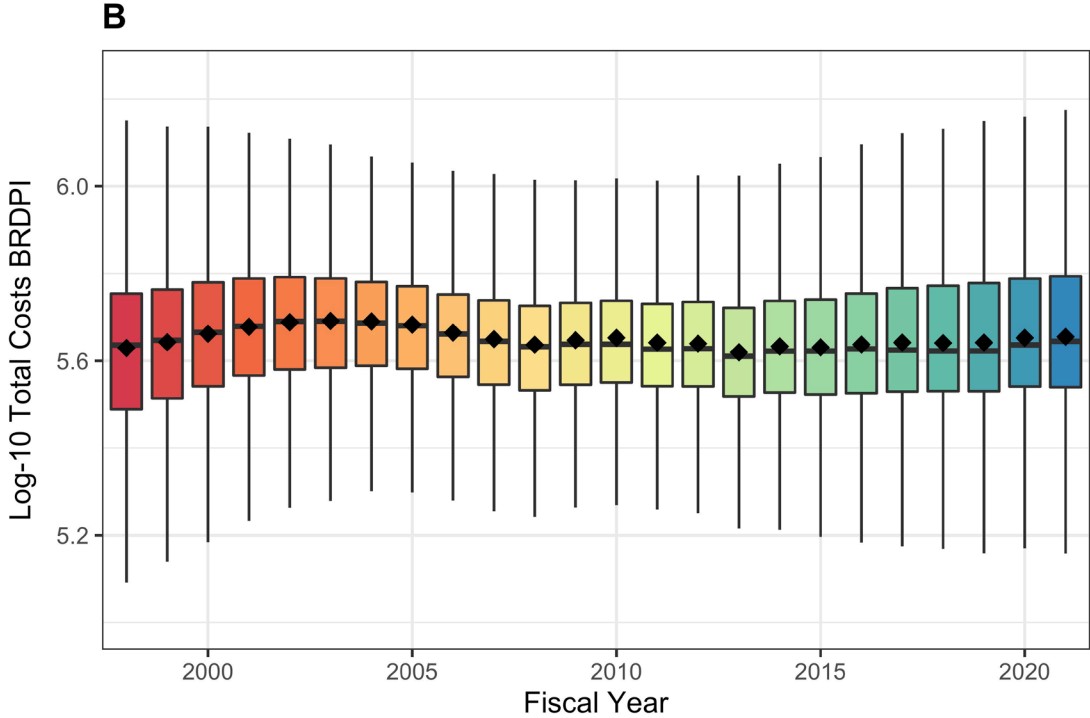

**Appendix 1—figure 2.** Box plot distributions of total costs (panel A) and log-transformed costs (panel B) for NIH-funded RPGs, FY1998 to FY2021. The diamonds refer to mean values. All costs are FY2021 BRDPI-adjusted.

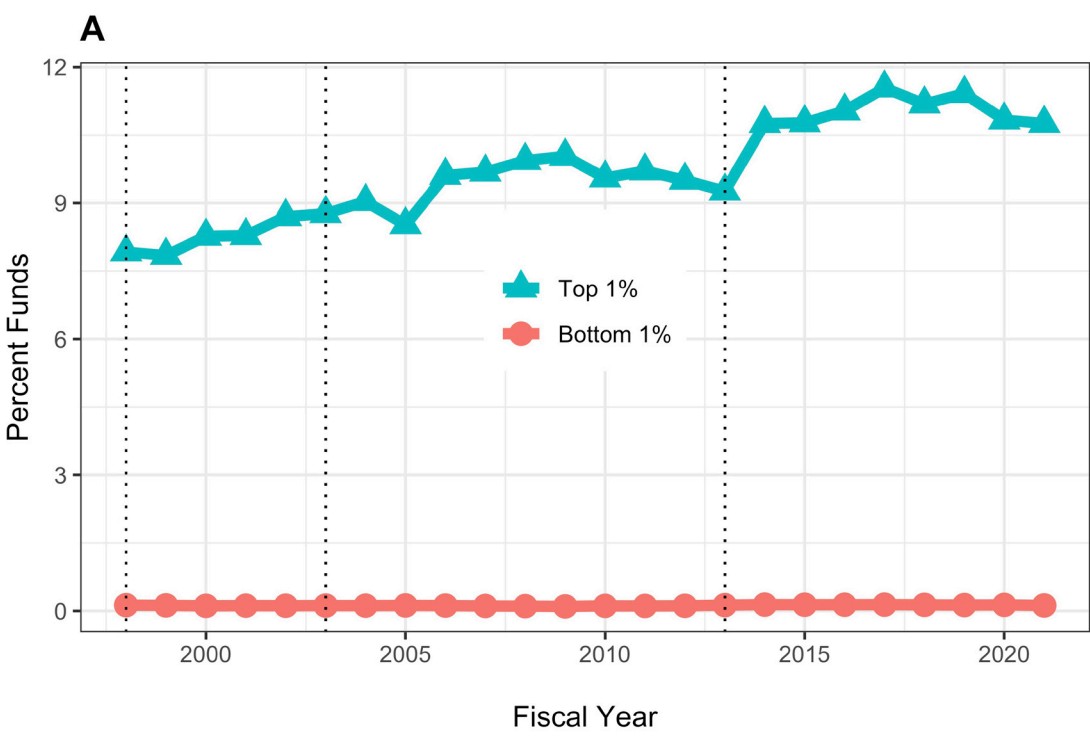

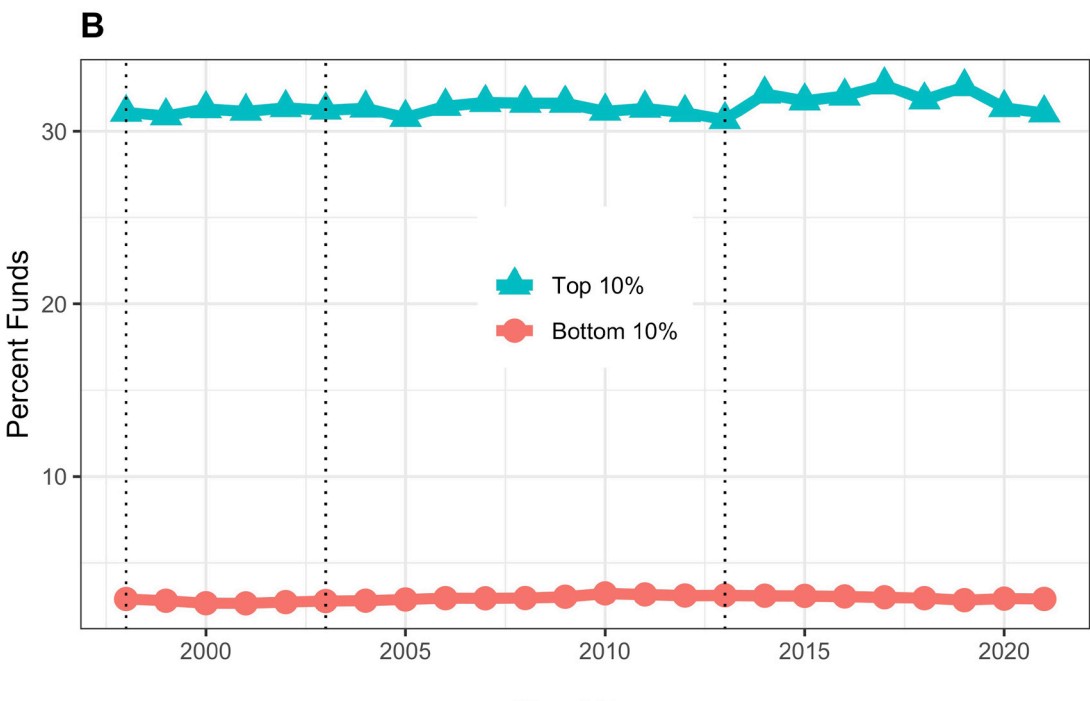

**Appendix 1—figure 3.** Distribution of total funds to the top and bottom centiles (**A**) and deciles (**B**) of NIH-funded RPGs, FY1998 to FY2021. The vertical dotted lines refer to the beginning and end of the NIH budget-doubling (FY1998 and FY2003) and the year of budget sequestration (FY2013).

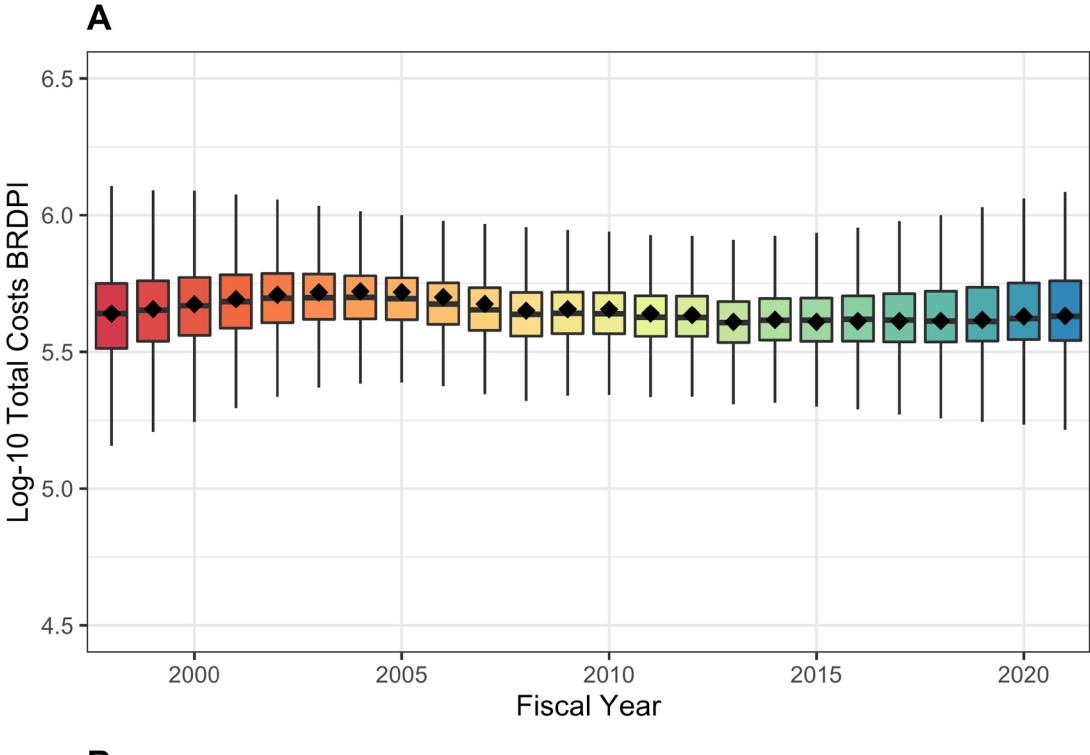

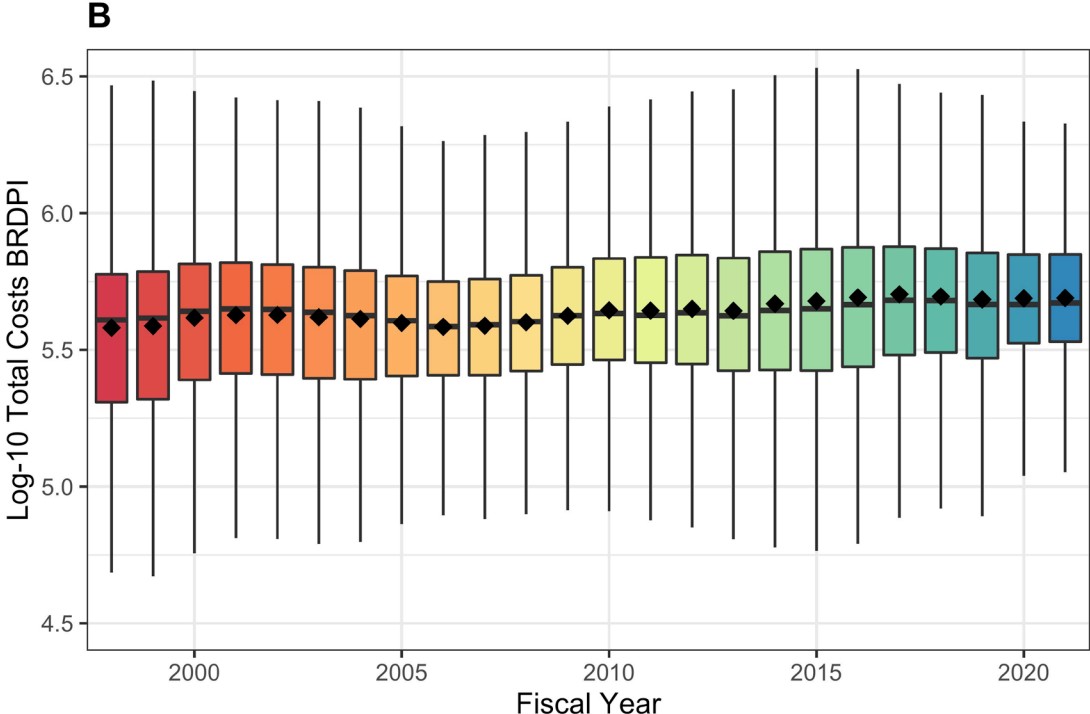

**Appendix 1—figure 4.** Box plot distributions of log-transformed costs for unsolicited (panel A) and solicited (panel B) NIH-funded RPGs, FY1998 to FY2021. The diamonds refer to mean values. Y-axes are similarly scaled for easier comparison. All costs are FY2021 BRDPI-adjusted.

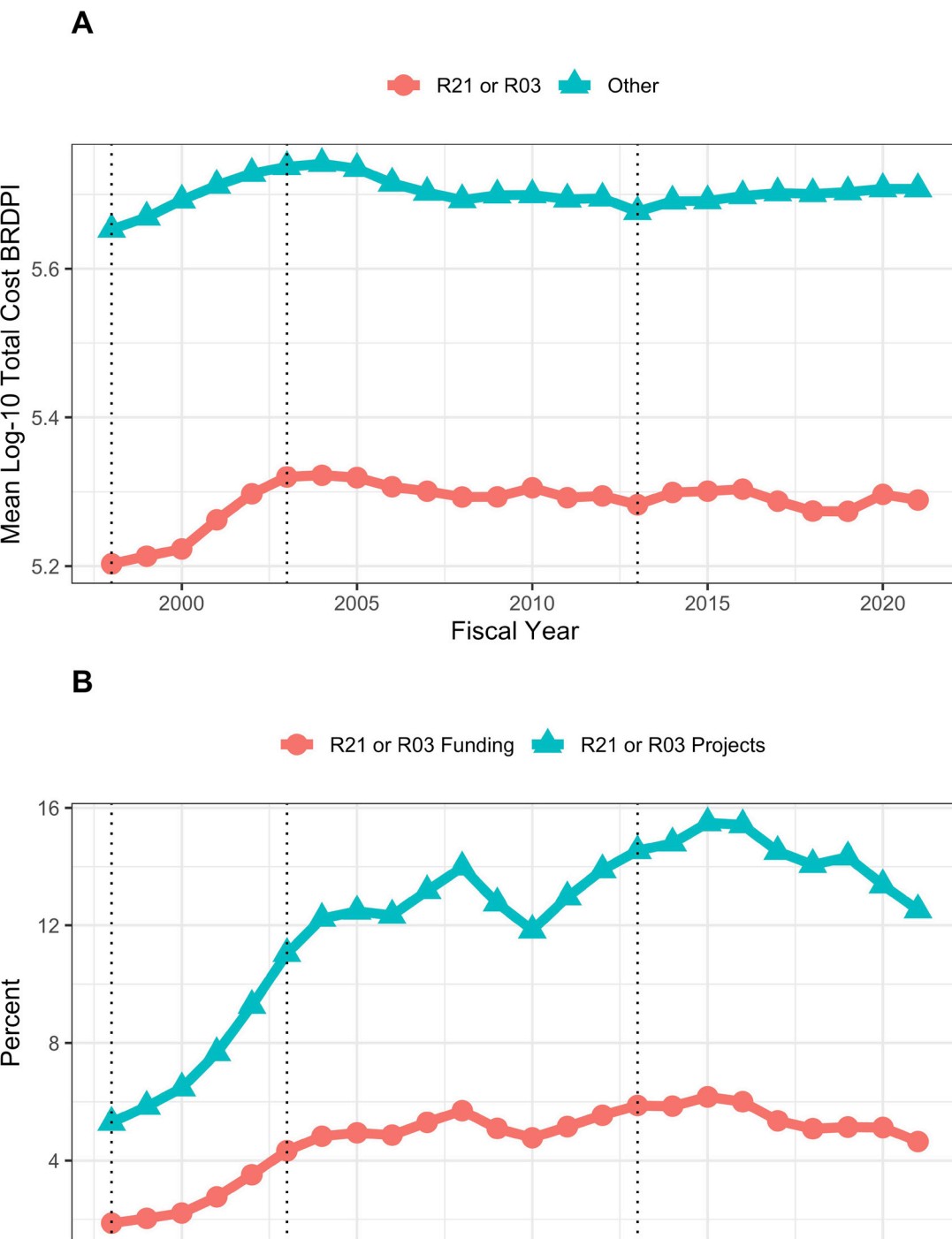

**Appendix 1—figure 5.** Trends in R21 and R03 projects and funding FY1998 to FY2021. (**A**): Costs according to R21 and R03 projects versus of all others. All costs are FY2021 BRDPI-adjusted. (**B**): Percent of RPG projects and percent of RPG funding going to R21 and R03 projects. The vertical dotted lines refer to the beginning and end of the NIH budget-doubling (FY1998 and FY2003) and the year of budget sequestration (FY2013).

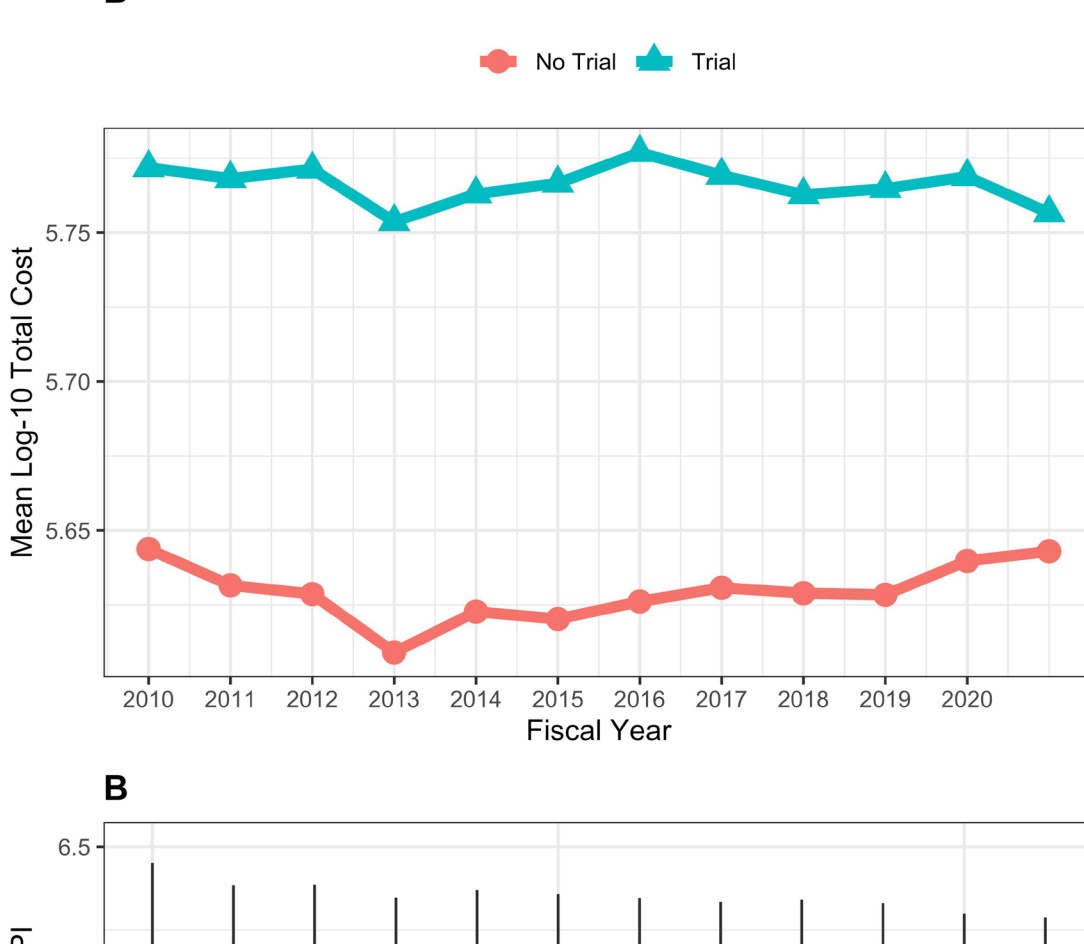

**Appendix 1—figure 6.** Trends in costs for RPGs involving and not involving clinical trials FY2010 to FY2021. (**A**): Log-transformed costs according to clinical trials for NIH-funded RPGs, FY2010 to FY2021. (**B**): Box-plot distributions of log-transformed total costs for NIH-funded RPGs supporting clinical trials, FY2010 to FY2021. All costs are FY2021 BRDPI-adjusted.

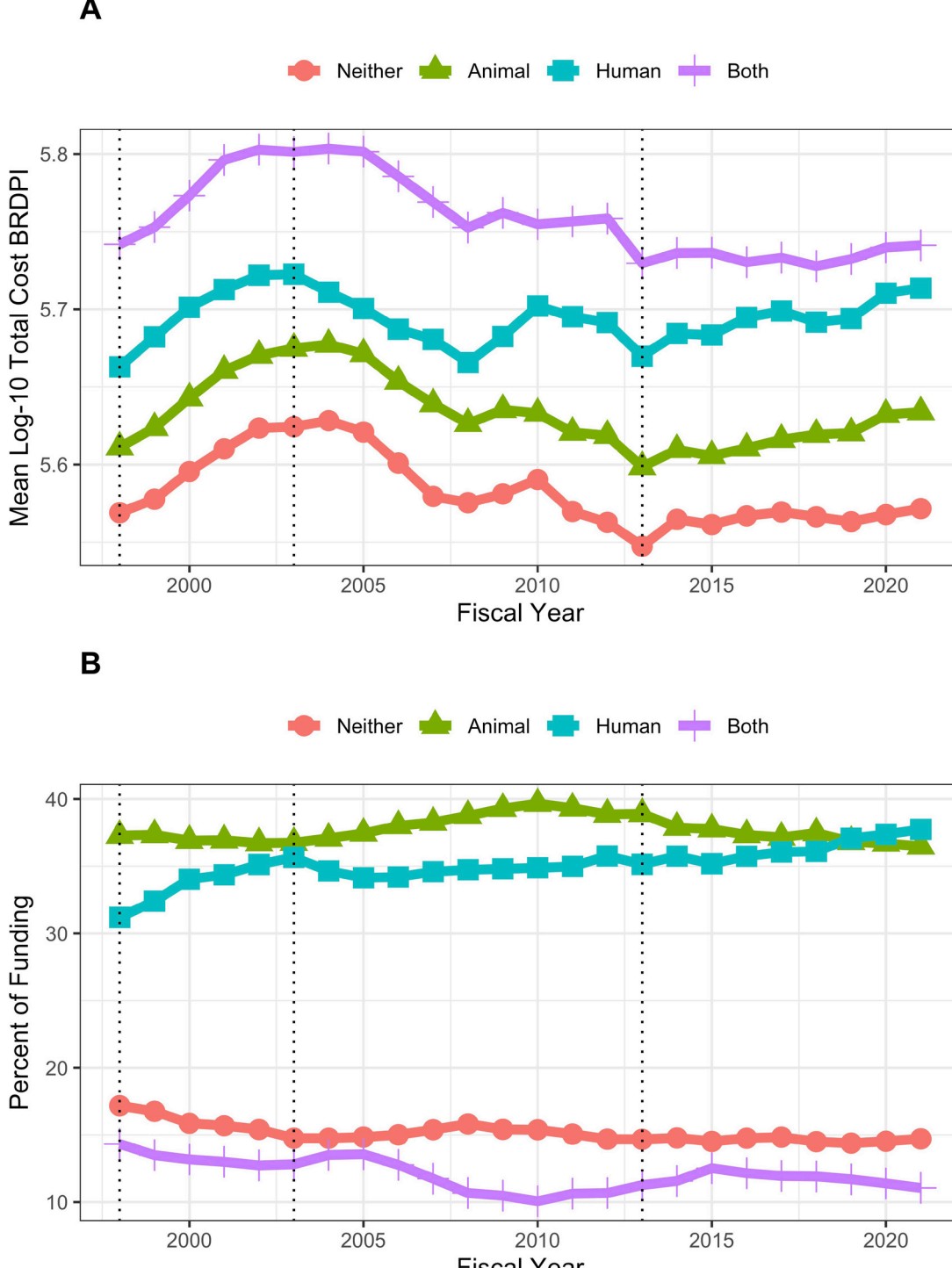

**Appendix 1—figure 7.** Trends in funding and costs for RPGs according to whether projects did or did not involve animal models and / or human participants FY1998 to FY2021. (**A**): Log-transformed costs of NIH-funded RPGs according to involvement of human participants and/or animal models, FY1998 to FY2021. All costs are FY2021 BRDPI-adjusted. (**B**): Proportion of RPG-funding going to different types of projects according to involvement of human participants and/or animal models, FY1998 to FY2021. The vertical dotted lines refer to the beginning and end of the NIH budget-doubling (FY1998 and FY2003) and the year of budget sequestration (FY2013).

