## [Editor Report]

Costs for NIH supported research go up each year and it is important to understand whether those costs are greater than those due to overall inflation which recently is rising more than expected. In this paper Lauer and colleagues report that prior to 2012 NIH costs exceeded inflation, but over the last ten years real NIH costs, match inflationary increases, due in part to salary caps on investigators. More of the funded NIH research during this time period is supporting solicited projects.

---

## [Decision Letter]

**Decision letter after peer review:**

Thank you for submitting your article "National Institutes of Health Research Project Grant Inflation 1998 to 2021" for consideration by *eLife*. Your article has been reviewed by 3 peer reviewers, including Clifford J Rosen as Reviewing Editor and Reviewer #1, and the evaluation has been overseen by a Senior Editor. The following individual involved in review of your submission has agreed to reveal their identity: Mark Peifer (Reviewer #2).

All the reviewers agree this is a timely manuscript. But there were several issues that need further delineation and clarification. These can be summarized below; more detail about concerns are noted in the three reviews.

Essential revisions:

1) Clarification in the text to help the reader understand some of the terminology and methodology, rather than just a series of graphs that dominate the manuscript.

2) There needs to be a clearer delineation of solicited grants and what their implications are compared to RO1s, particularly because the median grant cost is now much higher due to these grants.

3) Since personnel costs dominate NIH investigator grants the issue of salary caps and the implication for overall support of investigator needs more discussion.

4) Can the authors be more granular about the analysis relative to the majority of grants that are in the 250,000$ range and are well below the median level of 500k. In other words, are the results relevant for most RO1 investigators who continue to struggle with costs due to personnel and salary caps.

*Reviewer #1 (Recommendations for the authors):*

The authors provide a very comprehensive picture through modeling and graphic presentation of the relative pace at which NIH is able to maintain support for RPGs consistent with inflationary trajectories. Overall the paper is fairly well written, and the figures are illustrative, although some should probably be as supplemental so as not to overwhelm the reader.

The overall message is reassuring to investigators and institutions receiving NIH support.

There are several areas that require further elaboration:

1 – The abstract should be clearer in respect to methodology and overall message.

2 – What are the implications of larger funded, more clinically oriented projects relative to the fate of RO1s; this is paramount to investigators- particularly basic scientists.

3 – Personnel costs consume much of the budgetary costs of RPGs. Are your models able to capture that and if not, how does that affect the projections?

4 – Are indirect costs also keeping up with inflationary pressures? It might be embedded in the data but not clear to the reader.

5 – With the huge increase in inflation within the past year, readers would want to know if your modeling can predict how that might project for the future.

6 – Can the authors provide more detail on solicited projects since that seems to be consuming a greater proportion of the budgetary outlay?

7 – If NIH is supporting larger projects that are solicited, is there evidence that this support translates into greater productivity.

*Reviewer #3 (Recommendations for the authors):*

These comments are more "food for thought" for the research team.

This paper has been written at a very interesting time. Up until now, inflation has not been a problem for the US economy nor for NIH research. In fact, real funding per NIH project has fallen by about 10%. I find it both ironic and premature for the paper to be discussing Baumol's cost disease when prices per project are dropping (although the variance is changing).

For me, the issue is related to the salary cap. Salaries are the largest component of research costs, and NIH has maintained a cap that essentially has controlled prices, so it follows that the changes in NIH costs are going to be driven by composition effects. NIH effectively controlled labor costs from 2012 to present. The overall labor shortage in the US economy will ultimately have an impact on the cost of research funding given the reports of people being unwilling to take postdoctoral positions (see https://www.science.org/doi/pdf/10.1126/science.add6184 and https://www.statnews.com/2022/11/10/tipping-point-is-coming-unprecedented-exodus-of-young-life-scientists-shaking-up-academia/). I raised this issue in the public comments, and I would encourage the authors to address it head-on: the cost of doing research is going to increase rapidly in the next few years as outside opportunities to have a permanent job in industry paying twice the rate of an NIH postdoc abound.

This begs two questions: (A) What are the unintended consequences of this kind of these salary caps? Previous work by this research team has shown decreasing productivity of NIH funding (Lauer et al. 2017 https://www.biorxiv.org/content/10.1101/142554v2). Could this be the result of the reduction in the real value of each NIH grant? (B) NIH is going to have to raise the stipends for NSRA postdocs. The research team should mention that this will be likely in the future. Given the demand for skilled workers and cooling relations with China, fewer students will be going to graduate school and/or taking postdocs because the opportunity cost of foregone earnings has increased significantly. While we don't have a lot of evidence of Baumol's cost disease now, I bet five years from now that will have most certainly changed.

---

## [Author Response]

Essential revisions:1) Clarification in the text to help the reader understand some of the terminology and methodology, rather than just a series of graphs that dominate the manuscript.

The paper has been extensively rewritten, streamlined, and clarified, largely following the advice of the reviewers. See italicized text.

2) There needs to be a clearer delineation of solicited grants and what their implications are compared to RO1s, particularly because the median grant cost is now much higher due to these grants.

We have added material on solicited grants and R01-equivalent awards. See, for example, new Tables 2 and 4, and new Figure 4.

3) Since personnel costs dominate NIH investigator grants the issue of salary caps and the implication for overall support of investigator needs more discussion.

We have added new text on salary caps in the Discussion. See italicized text.

4) Can the authors be more granular about the analysis relative to the majority of grants that are in the 250,000$ range and are well below the median level of 500k. In other words, are the results relevant for most RO1 investigators who continue to struggle with costs due to personnel and salary caps.

We have added new material on R01-equivalent awards, including on the 250K and 500K levels. See new Tables 2 and 4.

Reviewer #1 (Recommendations for the authors):The authors provide a very comprehensive picture through modeling and graphic presentation of the relative pace at which NIH is able to maintain support for RPGs consistent with inflationary trajectories. Overall the paper is fairly well written, and the figures are illustrative, although some should probably be as supplemental so as not to overwhelm the reader.The overall message is reassuring to investigators and institutions receiving NIH support.There are several areas that require further elaboration:1 – The abstract should be clearer in respect to methodology and overall message.

We agree with the reviewer and revised the abstract accordingly. See italicized text.

2 – What are the implications of larger funded, more clinically oriented projects relative to the fate of RO1s; this is paramount to investigators- particularly basic scientists.

We agree with the question. We provide additional data on R01-equivalent awards. See two new tables (Table 2 and Table 4) and one new Figure (Figure 4) along with italicized text (lines 97-98, 108-111).

3 – Personnel costs consume much of the budgetary costs of RPGs. Are your models able to capture that and if not, how does that affect the projections?

We agree with the question. We have added text in the discussion (including limitations section) on personnel costs, including noting that our data are not as comprehensive as those of others (lines 265-267).

4 – Are indirect costs also keeping up with inflationary pressures? It might be embedded in the data but not clear to the reader.

We agree with the question. We provide additional data on indirect costs. See new Figure 3.

5 – With the huge increase in inflation within the past year, readers would want to know if your modeling can predict how that might project for the future.

Our paper is not designed to be a forecasting effort, but we have added displays for projected inflation indices in Figure 1, panel B.

6 – Can the authors provide more detail on solicited projects since that seems to be consuming a greater proportion of the budgetary outlay?

We agree with the question. We have provided more details on solicited and unsolicited projects in Tables 3 and 4. Table 4 is a wholly new Table on R01-equivalent awards.

7 – If NIH is supporting larger projects that are solicited, is there evidence that this support translates into greater productivity.

This is a good question, but beyond the scope of this paper. See italicized text (lines 269-271).

Reviewer #3 (Recommendations for the authors):These comments are more "food for thought" for the research team.This paper has been written at a very interesting time. Up until now, inflation has not been a problem for the US economy nor for NIH research. In fact, real funding per NIH project has fallen by about 10%. I find it both ironic and premature for the paper to be discussing Baumol's cost disease when prices per project are dropping (although the variance is changing).For me, the issue is related to the salary cap. Salaries are the largest component of research costs, and NIH has maintained a cap that essentially has controlled prices, so it follows that the changes in NIH costs are going to be driven by composition effects. NIH effectively controlled labor costs from 2012 to present. The overall labor shortage in the US economy will ultimately have an impact on the cost of research funding given the reports of people being unwilling to take postdoctoral positions (see https://www.science.org/doi/pdf/10.1126/science.add6184 and https://www.statnews.com/2022/11/10/tipping-point-is-coming-unprecedented-exodus-of-young-life-scientists-shaking-up-academia/). I raised this issue in the public comments, and I would encourage the authors to address it head-on: the cost of doing research is going to increase rapidly in the next few years as outside opportunities to have a permanent job in industry paying twice the rate of an NIH postdoc abound.This begs two questions: (A) What are the unintended consequences of this kind of these salary caps? Previous work by this research team has shown decreasing productivity of NIH funding (Lauer et al. 2017 https://www.biorxiv.org/content/10.1101/142554v2 ). Could this be the result of the reduction in the real value of each NIH grant? (B) NIH is going to have to raise the stipends for NSRA postdocs. The research team should mention that this will be likely in the future. Given the demand for skilled workers and cooling relations with China, fewer students will be going to graduate school and/or taking postdocs because the opportunity cost of foregone earnings has increased significantly. While we don't have a lot of evidence of Baumol's cost disease now, I bet five years from now that will have most certainly changed.

We agree with the reviewer. See new italicized text in the discussion (lines 189-203, 236-238).